# Improved cohesin HiChIP protocol and bioinformatic analysis for robust detection of chromatin loops and stripes

Karolina Buka [1,4] ✉, Zofia Parteka-Tojek [1,2,4], Abhishek Agarwal [1,4], Michał Denkiewicz [2], Sevastianos Korsak[1,2], Mateusz Chiliński[1,2,3], Krzysztof H. Banecki [1,2] & Dariusz Plewczynski [1,2] ✉

Chromosome Conformation Capture (3 C) methods, including Hi-C (a high-throughput variation of 3 C), detect pairwise interactions between DNA regions, enabling the reconstruction of chromatin architecture in the nucleus. HiChIP is a modification of the Hi-C experiment that includes a chromatin immunoprecipitation (ChIP) step, allowing genome-wide identification of chromatin contacts mediated by a protein of interest. In mammalian cells, cohesin protein complex is one of the major players in the establishment of chromatin loops. We present an improved cohesin HiChIP experimental protocol. Using comprehensive bioinformatic analysis, we show that a dual chromatin fixation method compared to the standard formaldehyde-only method, results in a substantially better signal-to-noise ratio, increased ChIP efficiency and improved detection of chromatin loops and architectural stripes. Additionally, we propose an automated pipeline called nf-HiChIP (https://github.com/SFGLab/hichip-nf-pipeline) for processing HiChIP samples starting from raw sequencing reads data and ending with a set of significant chromatin interactions (loops), which allows efficient and timely analysis of multiple samples in parallel, without requiring additional ChIP-seq experiments. Finally, using advanced approaches for biophysical modelling and stripe calling we generate accurate loop extrusion polymer models for a region of interest and provide a detailed picture of architectural stripes, respectively.

Genome folding is a complex process that efficiently packs roughly two-metre-long human DNA into a cell nucleus that is only a few micrometres in diameter. Importantly, it does not serve only structural purposes, but must be tightly coordinated with fundamental processes occurring in the nucleus, such as DNA replication, transcription, or DNA repair. Despite significant progress in 3D genomics in recent years, our understanding of the relationship between genome structure and function is still largely elusive and requires further investigation. There is a need for improved methods to detect chromatin contacts at different resolutions and for bioinformatic pipelines that comprehensively and efficiently analyse complex experimental data.

Studies of DNA structure in the nucleus using 3C-based technologies and microscopic imaging methods have shown that in mammals chromatin is folded at different levels of organisation in the nucleus. At the lower scale, interphase chromosomes occupy discrete regions within the nucleus called chromosome territories[1,2]. They are partitioned into alternating multi-megabase scale regions called compartments of two types, A and B, corresponding to euchromatin and heterochromatin, respectively[3]. Within the compartments, the 10 nm chromatin fibre folds into loops connecting elements that may be distant in the linear genome, such as promoters and enhancers. A subset of chromatin loops form larger structures called Topologically Associated Domains (TADs) or domains, which are sub-megabase regions that have stronger contacts within themselves than with other regions[4–6].

In mammalian cells, the major factors involved in loop formation are the evolutionarily conserved proteins cohesin and CCTC-binding factor (CTCF). Cohesin is a ring-shaped protein complex composed of four subunits: two structural maintenance of chromosomes (SMC) proteins, SMC1 and SMC3, and two non-SMC proteins: RAD21 and stromal antigen (SA) that in vertebrates comes in two versions STAG1 (SA1) or STAG2

[1]University of Warsaw, Centre of New Technologies, Laboratory of Functional and Structural Genomics, Warsaw, Poland. [2]Warsaw University of Technology, Faculty of Mathematics and Information Science, Laboratory of Bioinformatics and Computational Genomics, Warsaw, Poland. [3]Section for Computational and RNA Biology, Department of Biology, University of Copenhagen, Copenhagen, Denmark. [4]These authors contributed equally: Karolina Buka, Zofia Parteka-Tojek, Abhishek Agarwal. ✉e-mail: k.buka@cent.uw.edu.pl; d.plewczynski@cent.uw.edu.pl

(SA2, reviewed in[7]). Cohesin association with the DNA is dynamic: it encircles the DNA upon loading[8] and is able to slide along the DNA[9,10]. The sites of cohesin loading differ from the sites of its final genomic localisation[11]. The transcription process was shown to translocate cohesin along the DNA over long distances[10,12]. CTCF is a sequence-specific DNA binding protein that contains a highly conserved 11 zinc finger domain. CTCF, together with cohesin, is enriched at TAD boundaries[4,5] and at the majority of loop anchors[3,6]. In mammalian cells, cohesin colocalises with CTCF in most of its binding sites[13,14].

It was proposed[15–17] that loops are formed by a loop extrusion process, in which cohesin uses its ATP-dependent motor function to move along the DNA, causing active loop enlargement until it encounters a physical barrier, such as the CTCF protein, or until it dissociates from chromatin[18–20]. By stopping the extrusion process, CTCF acts as a chromatin loop anchor. It restricts the contacts occurring within the TADs and prevents them from crossing their borders. During the extrusion process promoters can be brought into contact with long clusters of enhancers[16].

3 C techniques, including Hi-C, detect interactions between different DNA regions, enabling the reconstruction of the chromatin architecture in the nucleus (reviewed in[21]). A standard Hi-C procedure involves chromatin structure fixation, restriction enzyme digestion, end repair, and ligation of fragments localised close to each other in the nuclear space. The DNA is then fragmented to generate linear chimeric fragments, which is followed by genomic library preparation, high-throughput sequencing, and bioinformatic analysis. Modifications of Hi-C experiments, such as HiChIP[22], PLAC-seq[23], similar to the ChIA-PET[24] method introduced in 2009, use a chromatin immunoprecipitation (ChIP) step, for genome-wide detection of interactions mediated by a specific protein. In the HiChIP approach, ChIP with the antibody of interest is performed after the Hi-C ligation step (Fig. 1A). An important advantage of the HiChIP approach over the standard Hi-C is that a higher resolution can be achieved with a lower sequencing depth.

HiChIP experiments require a extensive computational analysis to determine chromatin structure features such as: (a) HiChIP peaks detected from the signal of aligned sequencing reads, (b) chromatin interaction matrices showing all contacts detected in a given sample, (c) significant DNA interactions (loops) corresponding to the most frequent chromatin interactions present in the cell population, and finally (d) architectural stripes appearing on the contact matrices as horizontal or vertical lines[16]. Similar to loops and TADS, stripes are the population average representation of loop extrusion as detected by the 3C-based methods, where one loop (stripe) anchor is held in place, and the other loop anchor moves through the stripe domain until the loop is fully extruded[25,26].

Dedicated methods have been developed to analyse specific 3 C experiments, such as Juicer[27] for Hi-C, MAPS[28] for HiChIP and PLAC-seq data, and ChIA-PIPE[29] for ChIA-PET. In this study, we tested all three pipelines to analyse and compare our improved HiChIP protocol with publicly available HiChIP datasets. We also used two different stripe-calling algorithms: Stripenn[30], a state-of-the art tool operating on contact matrices, and the novel tool called gStripe, which is specifically designed to analyse sparse data using graph methods (see Methods for a description).

We observed that the quality of the cohesin HiChIP data is significantly lower compared to the CTCF HiChIP. This could be due to the more dynamic, mobile nature of the cohesin complex compared to CTCF, as discussed above. In this study, we present a modified version of the cohesin HiChIP protocol that significantly improves signal-to-noise ratio and loop detection efficiency. Importantly, we propose an automated pipeline "nf-HiChIP", based on the Nextflow system[31], for efficient and timely analysis of HiChIP samples starting from raw sequencing reads and ending with a set of significant chromatin interactions (loops). This pipeline can be easily deployed on a local workstation, a high performance computing system, or cloud-based computing environments. Finally, using high-quality cohesin HiChIP data obtained with improved protocol and a novel biophysical modelling approach for loop extrusion[32], we simulated dynamic 3D models of loop extrusion for a region of interest.

## Results

### Standard cohesin HiChIP protocol yields results with low signal-to-noise ratio

This study primarily investigates human lymphoblastoid cell lines (LCLs) from the 1000 Genome Project[33]. We observed that publicly available cohesin (SMC1) HiChIP for the GM12878 LCL[22] exhibits a much lower signal-to-noise ratio than the CTCF HiChIP performed for GM12878[34] and HG00731 (this study) LCLs (Fig. 1B, lanes 2, 6, 7; Supplementary Fig. 1A). A similar low signal-to-noise ratio was also observed in cohesin (SMC1) HiChIP experiments performed on the REC-1 and HCC1599 cell lines (ref. 35, Fig. 1B, lanes 3 and 4). Visual comparison between cohesin HiChIP and cohesin (SA1) ChIP-seq[36] signal in GM12878 further suggests limited enrichment of sequencing reads around cohesin ChIP-seq peaks in HiChIP experiments (Fig. 1B, lanes 2 and 5). We used SA1 ChIP-seq dataset[37] for comparative analysis because of two reasons: (1) SA1 ChIP-seq data recovered over 90% of the SMC3 ChIP-seq peaks[38] used for the cohesin HiChIP analysis by Mumbach et al.[22], while also identifying nearly 30000 additional binding sites (Supplementary Fig. 1B), and (2) the corresponding CTCF ChIP-seq was available for the same experimental setup enhancing the reliability of our comparative analyses by reducing potential batch effects. The summary of the HiChIP and ChIP-seq experiments used in this study is provided in Supplementary Table 1.

### Dual cross-linking protocol improves cohesin ChIP efficiency and enables successful detection of cohesin HiChIP peaks

To improve the quality of the cohesin HiChIP experiment, we tested a modified HiChIP protocol (Supplementary Fig. 1C) that includes an additional cross-linking agent to formaldehyde (FA) - ethylene glycol bis(succinimidyl succinate) (EGS), which previously improved signal-to-noise ratio in ChIP experiments[39,40]. The dual cross-linking protocol was also used for the ChIA-PET experiments[37,41], and it was recently shown to improve the quality of the Hi-C experiment[42]. For clarity, we have referred to the modified HiChIP protocol as FA-EGS HiChIP or dual cross-linking HiChIP (dcHiChIP) to distinguish it from the standard protocol. We performed cohesin FA-EGS HiChIP experiments for the HG00731 LCL in two biological replicates. By visual inspection in the genome browser and using the Pearson correlation coefficient, we confirmed a high reproducibility between two replicates for the following experiments: (1) cohesin FA-EGS HiChIP for HG00731, (2) CTCF HiChIP for HG00731, (3) cohesin[22] and (4) CTCF[34] HiChIP for GM12878 (Supplementary Fig. 1A and D). Therefore, we pulled both replicates for further analysis. Cohesin HiChIP experiments in REC-1 and HCC1599 cell lines[35] were performed in one replicate and, as such, were used in the following analysis.

Genome browser inspection revealed a higher signal-to-noise ratio in cohesin FA-EGS HiChIP compared to cohesin FA HiChIP experiments, achieving comparable signal-to-noise levels to CTCF HiChIP data (Fig. 1B, Supplementary Fig. 1A). To further assess the quality of the dcHiChIP data, we also compared it with cohesin and CTCF ChIP-seq experiments for GM12878 LCL, as ChIP-seq data were not available for HG00731 LCL. Analysis using the Pearson correlation coefficient (Fig. 1C) showed the strongest correlations between cohesin and CTCF HiChIP or cohesin and CTCF ChIP-seq data from the same cell line (GM12878 and HG00731). Interestingly, HG00731 cohesin FA-EGS HiChIP correlated better with GM12878 cohesin ChIP-seq ($r = 0.70$) than with GM12878 cohesin HiChIP ($r = 0.46$), despite differences in LCL sources. Moderate correlation was also found between HG00731 cohesin dcHiChIP and GM12878 cohesin HiChIP ($r = 0.46$), while lower correlations were observed with REC-1 and HCC1599 cohesin HiChIP (0.24 and 0.14, respectively). These findings suggest that dual cross-linking improves cohesin HiChIP signal quality. Limited correlation between cohesin dcHiChIP and standard HiChIP is likely due to lower signal-to-noise in the latter.

To better understand the difference between dcHiChIP and standard cohesin HiChIP we performed peak calling using the MACS3 algorithm. As no input sample is generated in the HiChIP experiment, the MACS3 function for calculating the local background of the HiChIP sample ($\lambda_{local}$)

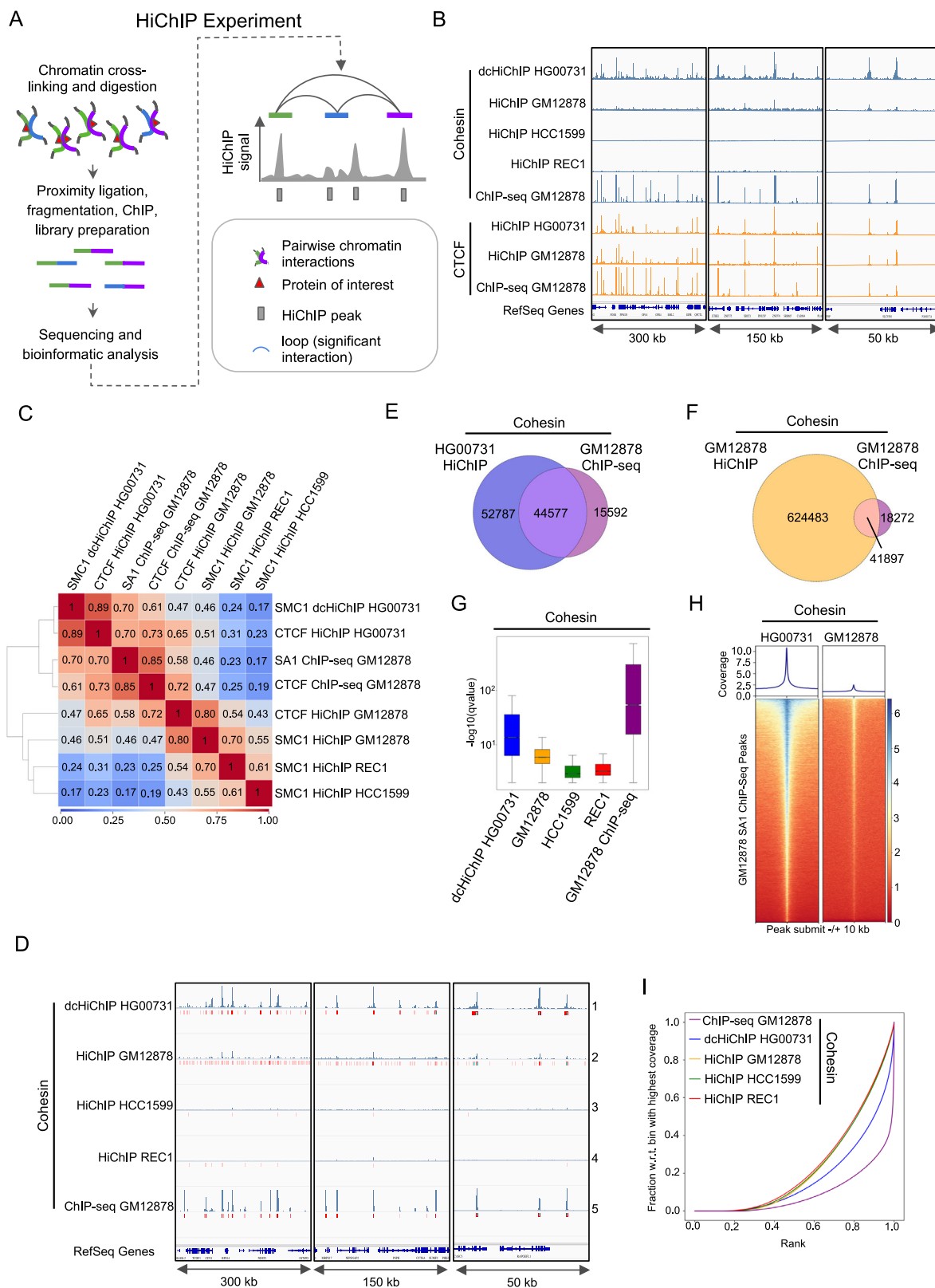

was used for peak enrichment analysis. The high overlap between peaks, called in separate replicates further confirmed the reproducibility of the HiChIP experiments (Supplementary Fig. 2). Over 600000 and nearly 100000 peaks were detected for cohesin HiChIP (GM12878) and FA-EGS HiChIP (HG00731), respectively. For the cohesin HiChIP experiment performed in REC-1 and HCC1599 a much lower number of peaks was

found (34,379 and 21,281, respectively). For comparison, MACS3 algorithm detected for cohesin and CTCF ChIP-seqs in GM12878 LCL ~60,000 and 67,000 peaks, respectively (Supplementary Table 2). We attribute the difference in peak counts between the HiChIP and ChIP-seq experiments to the biases of the 3 C step, which is not present in the regular ChIP-seq experiment. We were surprised by the high number of peaks detected for

**Fig. 1 | FA-EGS crosslinking improves the quality of the cohesin HiChIP experiment. A** Schematic of the HiChIP experiment and its data representation. **B** IGV browser coverage tracks of the HiChIP and ChIP-seq experiments analysed in this study. The following example genomic regions are shown: (1) 300 kb window - chr19:45,417,401-45,717,713; (2) 150 kb window - chr19:43,508,033-43,658,922; (3) 50 kb window - chr17:49,677,416-49,727,883. **C** Heatmap showing Pearson correlation coefficient between the samples processed in this study. **D** IGV browser coverage tracks and called peaks for the indicated cohesin HiChIP (SMC1) and ChIP-seq (SA1) experiments. The following example genomic regions are shown: (1) 300 kb window - chr5:150,332,698-150,635,698; (2) 150 kb window - chr7:55,939,000-56,089,000; (3) 50 kb window - chr17:40,160,000-40,210,000. **E** Venn diagrams showing common peaks between SMC1 HiChIP HG00731 and

SA1 ChIP-seq GM12878. **F** Venn diagrams showing common peaks between SMC1 HiChIP GM12878 and SA1 ChIP-seq GM12878. **G** Heatmap showing the distribution of SMC1 FA-EGS HiChIP HG00731 reads and SMC1 HiChIP GM12878 reads within a 20 kb window centred on the cohesin (SA1) ChIP-seq GM128787 peaks. **H** Distribution of -log(10) qvalue of peaks called by MACS3 in the indicated cohesin HiChIP and ChIP-seq samples. The horizontal red line inside each boxplot indicates the median. The bottom edge of the box marks the first quartile, and the top edge marks the third quartile and the whiskers extend to the minimum and maximum values, excluding outliers, which are not shown. **I** Fingerprint analysis plot (45) showing the comparison of ChIP enrichment in the indicated cohesin HiChIP and ChIP-seq samples.

cohesin HiChIP in GM12878[22] despite a rather low signal-to-noise ratio. Indeed, the genome browser view shows that some of the peaks are located in the regions that are not very distinct from the noise (Fig. 1D, lane 2). This was not the case for the FA-EGS HiChIP protocol, where peaks are distinct from the background noise (Fig. 1, lane 1). We hypothesise that the combination of the low signal-to-noise ratio of the data and the lack of input control may be a reason why the MACS algorithm detected so many peaks in this particular HiChIP experiment.

Importantly, 74% of the cohesin ChIP-seq peaks overlapped with the cohesin FA-EGS HiChIP peaks (Fig. 1E). We consider this a significant overlap taking into account that the experiment was performed in different LCLs (GM12878 vs. HG00731) and using different experimental protocols and antibodies targeting different cohesin subunits (SA1 vs. SMC1). For comparison, the overlap between cohesin HiChIP and ChIP-seq peaks for the GM12878 cell line was slightly lower (70%), although data were derived from the same cell line and that the number of HiChIP peaks was much higher (Fig. 1F). From this analysis we conclude that SMC1 dcHiChIP accurately detects cohesin binding sites. Moreover, this analysis indicates that when HiChIP data have a high signal-to-noise ratio, as is the case for the cohesin FA-EGS HiChIP, the lack of input control is not an obstacle to reliable HiChIP peak detection by the MACS peak calling algorithm.

To verify the influence of dual cross-linking on ChIP step efficiency in the HiChIP protocol, we analysed the enrichment of cohesin (SMC1) GM12878 HiChIP and HG00731 FA-EGS HiChIP reads around cohesin (SA1) GM12878 ChIP-seq peaks[36]. Strikingly, the enrichment of cohesin HG00731 FA-EGS HiChIP reads around GM12878 cohesin ChIP-seq peaks was substantially higher compared to GM12878 cohesin HiChIP (Fig. 1G). Furthermore, the significance levels of the set of peaks measured using the q-value parameter (MACS3) showed the highest score for FA-EGS HiChIP compared to other HiChIP samples (Fig. 1H). We also performed Fingerprint analysis[43,44]. According to this tool the better the quality of the ChIP sample is, the smaller the area under the curve (AUC) and the higher the elbow point it presents. As a reference we used the cohesin ChIP-seq experiment which, as expected, showed the highest quality. The FA-EGS HiChIP showed the best quality among the cohesin HiChIP examined (Fig. 1I). These results indicate that the efficiency of the ChIP step in the cohesin FA HiChIP protocol is limited and that the dual cross-linking step improves it. Our results confirm that HiChIP provides reliable information about cohesin binding to DNA, as previously reported[22]. We conclude that FA-EGS cross-linking improves the signal-to-noise ratio, increases the efficiency of ChIP step in cohesin HiChIP experiment and enables reliable detection of HiChIP peaks.

## Dual cross-linking HiChIP protocol improves detection of cohesin-mediated loops

In order to detect loops (significant interactions) connecting two distant genomic fragments (anchors), we used three independent algorithms: (1) nf-HiChIP and (2) ChIA-PIPE[29], tools specifically designed to process experiments targeting protein-mediated interactions (such as HiChIP, PLAC-seq or ChIA-PET) as well as, (3) Juicer[27], an algorithm originally developed for Hi-C samples. The first algorithm, nf-HiChIP, is based on MAPS[28]—an algorithm used for loop calling - which takes as input mapped,

paired and filtered reads, as well as the binding site information of the protein of interest (usually ChIP-seq experiment). The nf-HiChIP extends the algorithm by allowing the input to be raw sequences, calculating all the necessary preprocessing steps automatically. Identified interactions have at least one anchor that is enriched in a given protein factor. Juicer, on the other hand, generates interaction matrices (contact maps) which are then processed by a loop-calling algorithm (HiCCUPS). It identifies interactions by detecting groups of pixels representing the most frequent interactions, visible as "dots" on the contact maps. Therefore, loop detection by Juicer is independent of ChIP-seq peaks. ChIA-PIPE takes paired-end reads and provides clusters of the paired-end tags (PETs). It then merges overlapping PETs to determine the PET counts of potential chromatin contacts (or "looping") frequency between two anchor loci involved in chromatin interaction to estimate the strength of the loop.

We developed a multipurpose, Nextflow-based[31] pipeline called nf-HiChIP (https://github.com/SFGLab/nf-hichip) designed for NGS sequencing data analysis which performs (i) peak calling for ChIP-Seq and HiChIP data (ii) loop calling (i.e., identification of significant contacts) from HiChIP data and (iv) 2D contact matrix (i.e. hic file) (Supplementary Fig. 3). The nf-HiChIP pipeline is compatible with multiple samples and replicates, streamlining the analysis of large datasets through automation. Implemented in Python3[45] and using Nextflow, the pipeline excels at managing complex data workflows. Nextflow facilitates task scheduling and dependency resolution while providing robust error handling mechanisms—essential for large-scale analyses. The pipeline is structured into well-defined tasks, allowing for efficient error correction and resumption of the process from the last successfully completed task, eliminating the need to restart the analysis from scratch. The use of the pipeline is straightforward—installing can be done using 3 commands as shown in Nextflow documentation, and running HiChIP pipeline requires a sample design file. The rest of the workflow is fully automated, so the only input required is a.csv file describing the experiment and fastq files.

In our pipeline, in the loop calling step, we decided to use HiChIP peaks instead of ChIP-seq peaks. We reasoned that this solution would be particularly advantageous when the (1) ChIP-seq data is not available for the cell line and/or antibody of interest, (2) the available ChIP-seq data is of poor quality, or (3) the particular cell line has a slow proliferation rate and it is difficult to grow sufficient cell number for ChIP-seq and HiChIP experiments in parallel. To test whether this approach reliably detects chromatin interactions, we analysed the CTCF HiChIP (GM12878) sample, which has a high signal-to-noise ratio and for which CTCF ChIP-seq peaks are publicly available. We compared loops detected by the nf-HiChIP algorithm with CTCF ChIP-seq or CTCF HiChIP peaks and found that the overlap reached 98% (Supplementary Table 3). Additionally, we tested the similarity between nf-HiChIP loops (peak-dependent algorithm) and HiCCUPS loops, which are independent of peaks. For the HiCHIP samples analysed in this study, we have found that approximately 75% to 90% of nf-HiChIP loops were common with HiCCUPS loops (Supplementary Table 4, Supplementary Fig. 4) further validating the reliability of loops detected by nf-HiChIP using HiChIP peaks instead of ChIP-seq peaks. This confirms that when a HiChIP sample has a high signal-to-noise ratio, loop-calling analysis with nf-HiChIP can be performed using HiChIP

peaks from the same sample instead of using ChIP-seq peaks from additional experiments.

Visual inspection in the IGV and Juicebox genome browsers shows that all three algorithms detected more chromatin interactions in the FA-EGS HiChIP sample, compared to the other cohesin HiChIP samples analysed (Fig. 2A, B, Supplementary Figs. 5A, B, 6A, B and 7A, B). Indeed, in cohesin dcHiChIP (HG00731), the number of detected loops was at least twice as high as in the other cohesin HiChIP experiments analysed and was 80487, 43148 and 179516 for nf-HiChIP, HiCCUPS and ChIA-PIPE, respectively (Supplementary Table 5). The enrichment score of detected loops was calculated using aggregate peak analysis (APA,[6]) with nf-HiChIP, Juicer, and ChIA-PIPE loops based on interaction maps generated by Juicer software. In cohesin FA-EGS HiChIP, the aggregate loop strength was significantly better as compared to the remaining cohesin HiChIP experiments. It was comparable (in some cases even better) to the APA scores obtained for CTCF HiChIP experiments (Fig. 2C, Supplementary Figs. 5C and 6C). Therefore, in the cohesin HiChIP experiment, dual cross-linking improves loop detection in terms of loop number and strength in comparison to the FA cross-linking. To assess the robustness of our results across different loop-calling algorithms, we repeated our analyses using hichipper[46], FitHiChIP[47], and cLoops2[48]. We achieved a comparable number of identified interactions (Supplementary Table 5) and APA scores for the cohesin FA-EGS HiChIP experiment (Supplementary Fig. 8), indicating the reliability of our method across different loop-calling algorithms.

Visual examination of the loops in the genome browser identified from nf-HiChIP and HICCUPS with the fixed anchor size of 5 kb suggested that, despite the differences in the number of contacts detected (Supplementary Table 5), there is a high degree of concordance in contact localisation between cohesin FA-EGS HiChIP and other cohesin HiChIP experiments analysed (Fig. 2A, B, Supplementary Fig. 5A, B). Whereas in ChIA-PIPE the identified loops are with flexible anchor size and number of cohesin FA-EGS HiChIP loops was quite high as compared to other cohesin HiChIP samples (Supplementary Table 5) and ChIA-PIPE loops were more densely packed (Supplementary Fig. 6A, B). Indeed, using the nf-HiChIP pipeline, the loop overlap between cohesin FA-EGS HiChIP (HG00731) and cohesin HiChIP obtained for GM12878, REC-1 and HCC1599 was 95%, 80% and 85% respectively (Fig. 2D). We observed similar concordance in HiCCUPS interactions (Supplementary Fig. 5D). Within the interactions identified from ChIA-PIPE we identified similar behaviour but with lower overlap percentage with the short read sequencing samples HCC1599 and REC-1 (Supplementary Fig. 6D). This confirms the reliability of the improved HiChIP protocol and indicates that it allows for the generation of more complex maps of cohesin-mediated chromatin contacts than the standard one. The comparison between cohesin FA-EGS HiChIP and FA HiChIP shows that the general pattern of the sequencing signal on the interaction matrices is similar between experiments, but standard protocol produces a more diffused signal (Fig. 2B and Supplementary Figs. 5B, 6B and 7A, B).

Of note, not all peaks called from the raw sequencing data will be involved in chromatin interactions. Analysis of the proportion of HiChIP peaks that localise inside the loop anchors (3D peaks) detected by nf-HiChIP revealed that this percentage was the highest for cohesin FA-EGS HiChIP (51%), followed by cohesin HiChIP in HCC1599 and CTCF HiChIP in GM12878 (39% and 28% respectively). The lowest proportion of the 3D peaks within HiChIP anchors was observed for both CTCF and cohesin HiChIP peaks in GM12878 LCL and cohesin HiChIP in REC-1 which are 15%, 11% and 17% respectively (Supplementary Table 2). Interestingly, 3D peaks present higher cohesin or CTCF enrichment than peaks that are not involved in chromatin interactions (Supplementary Fig. 9A). As expected, the analysis of CTCF motif orientation within loop anchors revealed that CTCF motifs are positioned predominantly in convergent orientation for all the HiChIP experiments analysed (Supplementary Fig. 9B, Supplementary Table 6). Additionally, Chromatin loops were identified and categorized as Enhancer-Promoter (EP) or Promoter-Promoter (PP) across different cell lines using three loop-calling methods (Supplementary Table 7). Approximately 40–50% of the loops identified by

nf-HiChIP and HiCCUPS were classified as EP or PP, whereas the proportion was slightly lower for ChIA-PIPE (around 20–35%), likely due to the higher overall number of loops identified by this method. These findings provide deeper insights into the biological relevance of these loops.

## Dual cross-linking HiChIP protocol reveals architectural stripes in more detail

To assess the quality of the FA-EGS HiChIP data with respect to the detection of structures above the loop level, we perform the calling of architectural stripes. Stripe calling was performed on the cohesin HiChIP datasets using the Stripenn tool[30], which is based on image processing, and a novel method recently developed in our laboratory called gStripe, based on graph analysis (see Materials and Methods for details). Stripenn uses as input the heatmaps generated by Juicer, while gStripe operates on loops obtained in previous steps from either nf-HiChIP, HiCCUPS or ChIA-PIPE tools.

First, we compared how many stripes could be extracted from each dataset. The number of stripes called by Stripenn in the cohesin FA-EGS HiChIP sample (HG00731) is 8% lower than in the cohesin HiChIP (GM12878) sample (3603 vs 3908), but more than two and three times higher in comparison to REC-1 and HCC1599 cohesin HiChIP samples, respectively (Fig. 3A, Supplementary Table 8). On the other hand, gStripe consistently detected many more stripes in cohesin datasets derived from FA-EGS HiChIP, regardless of the loop-calling tool used. In particular, gStripe detected two to four times more stripes for cohesin FA-EGS HiChIP compared to standard cohesin HiChIP (GM12878) using nf-HiChIP, HiCCUPS and ChIA-PIPE input loops.

To get a more complete picture, we visually compared the stripe regions obtained from both protocols, and constructed pileup plots of the stripe domains in line with the methodology used in Stripenn[30]. The plots show the interaction matrix signal averaged over the stripe domains in each dataset (Supplementary Fig. 10). Crucially, we observed that the plots for cohesin FA-EGS HiChIP (HG00731) have sharper and more distinct stripe features than those obtained from cohesin FA HiChIP (GM12878), regardless of the stripe calling method. Moving further to visual inspection of individual stripes, we noticed, that often what appears as a smooth stripe signal in the standard HiChIP, often contains chains of distinct dots (loops) in FA-EGS HiChIP, (see e.g. the uppermost stripe in Fig. 3B or Supplementary Fig. 11A). Indeed, the Stripenn stripes in cohesin FA-EGS HiChIP (HG00731) contain 1.5, 1.8 and 2.6 times (for HiCCUPS, nf-HiChIP and ChIA-PIPE loops respectively) more loops per unit of area on the heatmap, than in cohesin FA HiChIP (GM12878, see Supplementary Table 9). As the image processing based Stripenn method expects a smooth signal, some of these stripes may be disturbed. However, they can be recovered by gStripe. This is consistent with the pileup plots, where the Stripenn stripes appear as wide, blurry flares, while gStripe stripes are thin and sharp. Likewise, it is in agreement with our preliminary observation that the FA-EGS HiChIP data appear to be more sensitive to changes in the parameters of Stripenn: within the same set of parameters of stripe calling, the ratio of the highest to lowest number of stripes called was 15.3 for HG00731 and only 2.5 for GM12878 (see Supplementary Table 9 for the full summary).

Next, we investigated how the stripes are situated in relation to each other, by calculating the proportion of stripes located within the domain of another, larger stripe (see Fig. 3B for an example). The stripe domain is defined as the range of coordinates from the stripe anchor to the end of the stripe. We found that a substantial proportion of the gStripe stripes are located within the domain of a larger stripe (between 57% when using nf-HiChIP loops, to up to 86% for ChIA-PIPE loops in cohesin FA-EGS HiChIP in HG00731 and between 38% and 70% for cohesin FA HiChIP in GM12878 respectively). We can therefore conclude that they often colocalise. On the other hand, the percentage of colocalized Stripenn stripes is lower: 21% for cohesin FA-EGS HiChIP (HG00731) and 31% for cohesin standard HiChIP (GM12878). In summary, the additional stripes detected by gStripe in cohesin dcHiChIP (HG00731) are smaller and more often colocalized within larger ones, especially when using a dense loop set such as

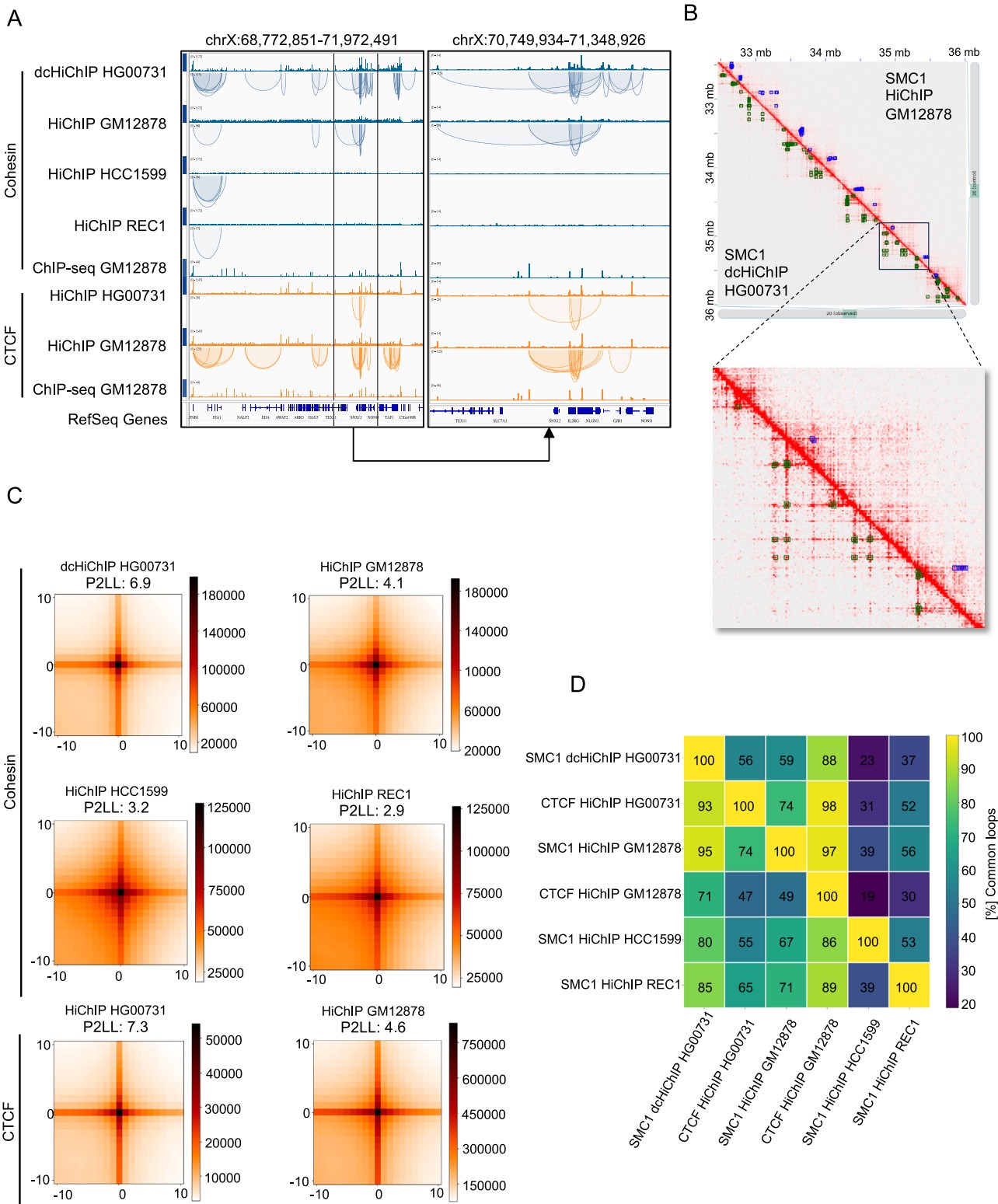

**Fig. 2 | FA-EGS crosslinking HiChIP protocol improves cohesin-mediated loop detection. A** IGV browser coverage tracks and nf-HiChIP loops of the indicated HiChIP samples and read coverage tracks for SA1 and CTCF GM12878 ChIP-seq experiments. **B** Juicebox interaction maps at 5 kb resolution of the example region for the indicated cohesin HiChIP samples. Loops (nf-HiChIP) are shown as green (SMC1 HiChIP HG00731) or blue (SMC1 HiChIP GM12878) rectangles. **C** APA analysis performed using loops identified by nf-HiChIP for the indicated cohesin and CTCF HiChIP samples. The APA score P2LL (peak to left lower corner) is the ratio of the central bin to the average of the lower left corner and indicates the strength of the loop. **D** Overlap between loops (with 15 kb tolerance) called by nf-HiChIP in the indicated datasets. A single cell in the matrix represents the percentage of loops in the row sample that overlap with one or more loops from the column sample. Note that the matrix is not symmetric, as one-to-many overlaps are possible.

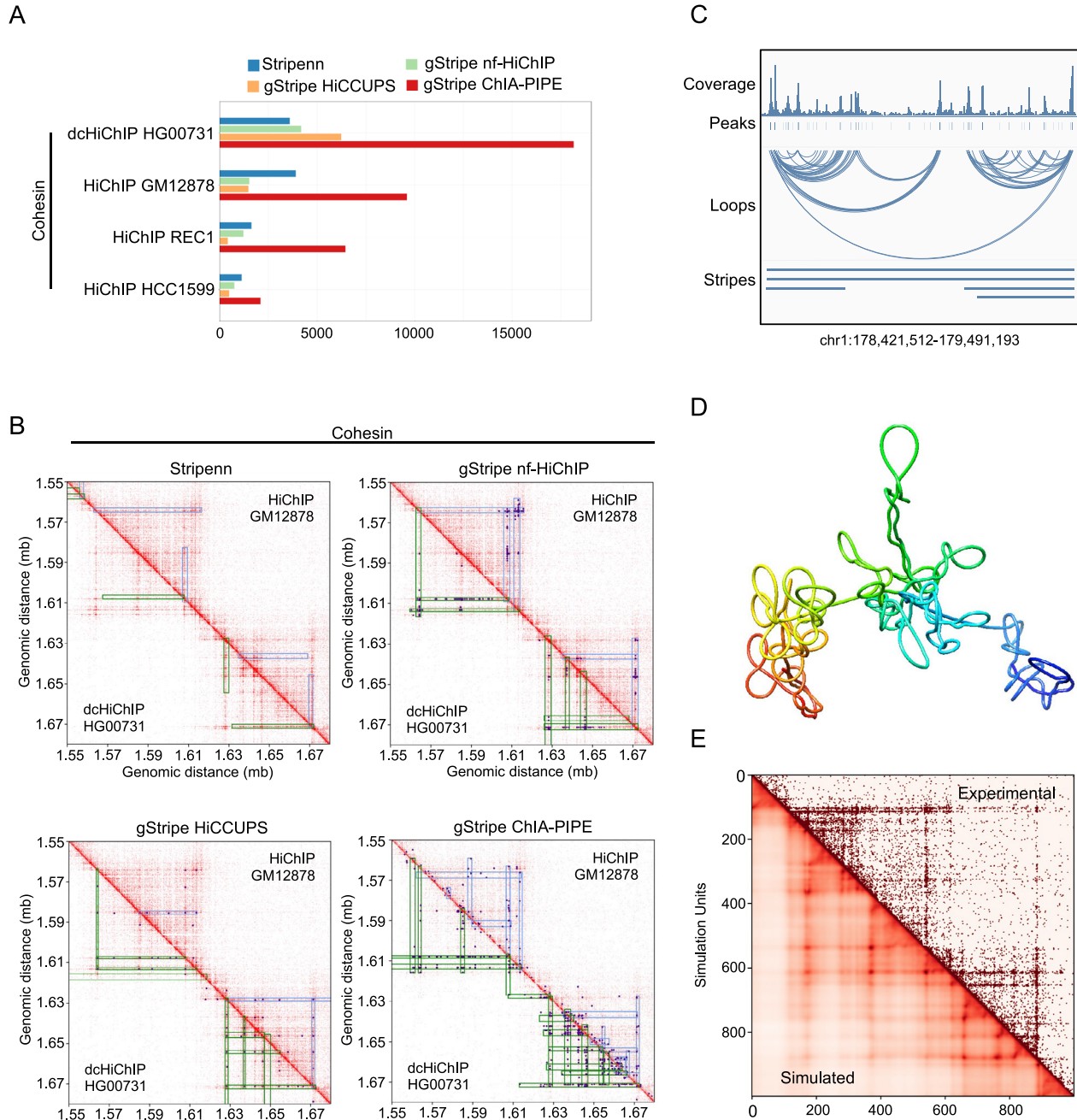

**Fig. 3 | FA-EGS crosslinking HiChIP protocol improves stripe detection and enables the building of accurate loop extrusion models. A** Number of stripes called by Stripenn and gStripe for the cohesin HiChIP datasets analysed in this study. gStripe results were calculated using nf-HiChIP, HiCCUPS or ChIA-PIPE loops as an input. **B** Juicebox interaction maps at 5 kb resolution and the stripes of the exemplary region (chr6: 15,550,000-16,850,000) for the cohesin HiChIP samples. Lower (below diagonal) and upper (above diagonal) corners present SMC1 FA-EGS HiChIP (HG00731) and SMC1 HiChIP (GM12878), respectively. Stripes were called by the algorithms indicated above the maps. Each stripe set is shown on top of data used to generate it: the interaction heatmap and, in the case of gStripe, the nf-HiChIP, HiCCUPS or ChIA-PIPE loop set (blue dots). **C** IGV browser view of the example region for SMC1 HG00731 FA-EGS HiChIP sample used for loop extrusion modelling. The first three tracks show the output of the nf-HiChIP pipeline: read coverage, peaks called by MACS3 and loops called by nf-HiChIP. The fourth track shows the stripes identified by gStripes (with the nf-HiChIP loop set). **D** Final structure obtained after running LoopSage simulation for the region and sample indicated in panel (**C**). **E** Experimental (above diagonal) versus simulated(below diagonal) heatmaps (at the 5 kb resolution) for the region and data shown in panel C.

the ChIA-PIPE, in comparison to standard cohesin HiChIP (Fig. 3B and Supplementary Fig. 11A).

Moreover, to assess the reliability of the stripe detection, we check how many stripes from one dataset can be found in other sets. We look both at the overlap of individual stripe anchors and whole stripe domains. In the cohesin FA-EGS HiChIP (HG00731) Stripenn found approximately 49% of the stripes and 74% of the stripe domains detected in GM12878, 52% stripes

and 78% domains from REC-1 HiChIP, and 43% stripes and 76% domains from HCC1599. For the gStripe algorithm, the degree of stripe overlap depends on the input loops set, but in general the overlap of stripes with the FA-EGS HiChIP (HG00731) was higher for cohesin HiChIP in GM12878 (e.g. 75% for nf-HiChIP loops) than for REC-1 and HCC1599 loops (e.g. 59% and 51% respectively, for nf-HiChIP). In the case of stripe domains, the overlap is very high: from 82% of HCC1599 domains recovered in HG00731

(using the HiCCUPS loop set) up to 98% of GM12878 domains (also for the HiCCUPS loops). The high overlap confirms the validity of the detected stripes.

Importantly, gStripe is able to recover a substantial number of GM12878 stripes (up to 88% when using HiCCUPS loop set) and stripe domains (98%, also for the HiCCUPS loop set) Since gStripe detected more stripes in FA-EGS HiChIP (HG00731), we can say that in this data, the set of visible stripes and stripe domains is extended, as compared to GM12878. A complete set of overlap statistics between stripe anchors and stripe domains are shown in Supplementary Fig. 11B and Supplementary Fig. 11C, respectively. Taken together, our results suggest that the FA-EGS HiChIP experiment provides similar quality of detection for stronger, more uniform stripes using an established image-driven method, Stripenn, while having an advantage in revealing finer, colocalized, stripes and uneven stripe structure when using a graph-based algorithm, gStripe.

Finally, we evaluate the suitability of the three loop callers (nf-HiChIP, HiCCUPS and ChIA-PIPE) for use with gStripe on FA-EGS HiChIP data. While ChIA-PIPE identifies the largest number of stripes, they frequently occupy overlapping domains (86% in the FA-EGS HiChIP sample vs. 57% for nf-HiChIP, see above), which may not always be desirable. In contrast, nf-HiChIP stripes exhibit the highest consistency across datasets, with 51%-75% stripes from GM12878 also present in HG00731, and they show the strongest pile-up plot signal among the gStripe results (Supplementary Fig. 10). Thus, while specific research goals may influence the choice of loop calling method, we generally recommend nf-HiChIP for generating input for gStripe, unless densely packed stripes in large numbers are needed, in which case ChIA-PIPE is a viable alternative.

## Dual cross-linking HiChIP protocol allows us to build accurate loop extrusion models

Finally, we aimed to generate biophysical models of the loop extrusion process to gain deeper insight into the role of cohesin dynamics in chromatin organization. By integrating biophysical modelling with experimental data, we aim to bridge the gap between observed chromatin interactions and the mechanisms that drive them. Usually, to simulate loop extrusion, stochastic models are used[15,25,49–51], where the cohesin proteins on the chromatin fibre perform one-dimensional random walk motion, and they may unload or load in different regions of chromatin stochastically. For the purpose of this work, we used our recently developed method, LoopSage[32], which combines stochastic simulation and molecular dynamics with OpenMM. In this model, cohesin rings play the role of loop extruders, and they follow a random walk diffusive motion, whereas CTCF acts as orientation-dependent barriers for its motion[17,52]. It also assumes that the position of CTCF proteins is more stable in comparison to cohesin complex[53,54].

Importantly, inspired by the report by Gibcus et al.[38], and publications of Samejima et al.[55], Hildebrand et al.[56], Dekker and Mirny (2024)[57] we have refined our original LoopSage method by introducing two different classes of loop extrusion protein complexes with different speeds. Specifically, we added a second population of faster-moving extruding factors (cohesin complexes) in addition to the slower-moving ones initially modelled. The fast-moving cohesin complexes are fewer in number but can form long-range loops more rapidly, capturing distant interactions observed in the experimental data. For a more detailed description of the biophysical modelling please refer to material and methods. This adjustment improved the alignment between our simulations and the experimental heatmaps, as demonstrated in the Supplementary Fig. 12A, B.

We performed a simulation for the SMC1 FA-EGS HiChIP sample (HG00731) for a region of chromosome 1 (Fig. 3C). Loop set from nf-HiChIP and the CTCF orientation at the loops anchors was taken as an input. Cohesin rings are then distributed according to their binding preferences. As the temperature decreases, we end up with a dynamical trajectory of 3D structures, where we assume that the final structure is the most representative one (Fig. 3D). We observe that the final polymer is condensed into two dominant clusters of loops, which correspond to the two squares in

the experimental and reconstructed simulation heatmaps (Fig. 3E). Comparison between the simulated heatmap, which is averaged over the ensembles of 3D structures and the experimental SMC1 dcHiChIP heatmap shows that loop and stripe patterns are well reconstructed (loop-strength correlation between experimental and simulated map is 98.4% and the overall heatmap correlation was 81%, Fig. 3E and Supplementary Fig. 12B), indicating that our biophysical assumptions were valid. Some discrepancies still arise due to the model limitations and its stochastic nature.

In conclusion, by integrating dcHiChIP data and refining LoopSage parameters, we verified assumptions such as cohesin diffusion, its stochastic binding and unbinding from the DNA. We also advanced the ability of LoopSage to simulate realistic 3D chromatin architecture. These results emphasise the importance of combining experimental data with computational modelling to improve our biophysical understanding of chromatin organisation.

## Discussion

In this study, using a comprehensive bioinformatics approach (summarised in the Fig. 4), we show that the cohesin HiChIP performed with the sequential (FA-EGS) cross-linking protocol improves the signal-to-noise ratio and ChIP efficiency in comparison to the standard one[22] and enhances the detection of chromatin loops and stripes in human lymphoblastoid cells. We have developed nf-HiChIP pipeline that combines the analytical approach designed for ChIP-seq data processing (mapping, filtering, peak calling, coverage tracks calculations) with HiChIP-specific analysis (MAPS pipeline[28]) to facilitate the user the comprehensive and timely analysis of many HiChIP datasets in parallel, without the need to perform additional ChIP-seq experiments. Furthermore we show that the data derived from the improved cohesin HiChIP protocol can be combined with the recently developed energy-based modelling approach LoopSage[32] to generate accurate biophysical models of the loop extrusion process.

Dual cross-linking was previously reported to improve the signal-to-noise ratio and increase the sensitivity of the ChIP assay in the case of proteins with hyperdynamic exchange with DNA or indirectly DNA-bound proteins[39,40,58]. Cohesin also exhibits a dynamic association with DNA. We report that dual cross-linking improves the quality of cohesin HiChIP, which subsequently results in higher sensitivity of the experiment compared to standard cohesin HiChIP. Importantly, our results suggest that the increased efficiency of the 3 C step of the protocol may be another factor contributing to the improved detection of 3D chromatin features such as loops and stripes. The use of dual cross-linking (EGS or DGS and FA) has been shown to reduce the experimental noise by decreasing the number of random Hi-C ligation events, leading to the improved loop detection[42,59]. Similarly, we observed more looping interactions in FA-EGS HiChIP, compared to cohesin FA HiChIP and these interactions were stronger. Importantly, the use of the FA-EGS HiChIP method allows the detection of 2-3 times more loops in human cells with 4-6.5 less sequencing depth compared to dual cross-linking Hi-C[42], indicating that the former method may be a more suitable and economical choice for studies focusing on the 3D chromatin structure at the resolution of loops. However, it is important to note that the influence of the cross-linking procedure on the quality of cohesin HiChIP quality might be cell type specific. For example, in the case of the undifferentiated cell line, human embryonic stem cells[60], a high signal-to-noise was obtained for the cohesin HiChIP experiment despite the use of only one cross-linking agent (FA) for fixation.

Our aim was to thoroughly investigate the results that can be obtained from HiChIP sample processing. Commonly used HiChIP data analysis tools like Juicer, HiC-Pro, and Fit-Hi-C, which were not specifically designed for HiChIP, may use normalisation and loop calling methods appropriate for Hi-C datasets. These methods may overlook the specific interactions in HiChIP. Unlike Hi-C, where a sparse interaction matrix is undesirable and is corrected using techniques such as Knight-Ruiz (KR) normalisation[6,27,61] to account for experimental bias, HiChIP interactions typically yield a sparser matrix. Using HiChIP-specific tools such as MAPS[28], we identified interactions that are undetectable when using KR-

**Fig. 4 | Systematic representation of the bioinformatic analysis and the pipelines used in this study.** Raw HiChIP data were processed using three independent pipelines: (1) Juicer[27], (2) nf-HiChIP, and (3) ChIA-PIPE[29]. The outputs generated from these pipelines were subsequently utilised as inputs for specialised analytical tools, including APA analysis, Stripenn[30] and gStripe for stripe detection, and LoopSage[32] for 3D genome modelling.

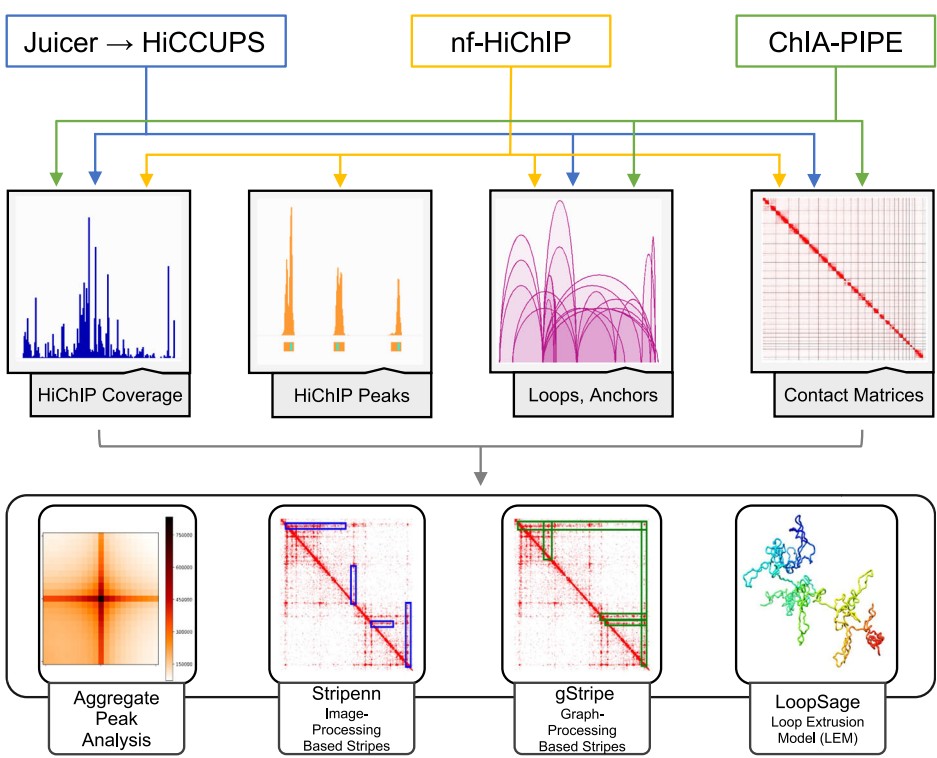

normalised interaction maps for loop calling. We also experimented with replacing KR-normalised maps with Vanilla Coverage (VC) normalisation[6] in the Juicer pipeline. Integrating VC normalisation, which unlike KR, does not assume a uniform distribution of interaction frequencies and thus preserves unique HiChIP-specific interactions, improved the consistency of our results. However, image-based loop calling tools such as HiCCUPS[27] may exclude closely positioned loops or those that start or end within the same matrix pixel, potentially indicating loop extrusion processes, that are important in the context of cohesin HiChIP data. These loops however, could be captured by direct processing of interaction pairs in HiChIP-focused approaches such as MAPS.

nf-HiChIP pipeline includes ChIP-seq-specific steps, including (1) a mapping approach that differs from the Hi-C mapping approach, (2) coverage track generation, and (3) peak calling that can be applied to both HiChIP and corresponding ChIP-seq samples, allowing dual analysis of HiChIP samples - both as ChIP-seq and HiChIP datasets. This dual approach provides more comprehensive outputs: peaks, coverage, and loops, from a single sample, increasing the data richness compared to traditional methods that require separate, manual analyses. This adaptability enhances the personalisation of analyses based on the experimental data available. Our results suggest that if the HiChIP experiment has a high signal-to-noise ratio, additional ChIP-seq is not required for subsequent data analysis. Such an approach may be particularly advantageous for projects investigating differences in 3D chromatin structure between many individuals or under various experimental treatment conditions in parallel (e.g. use of different sources of cellular stress). In such an experimental setup, each additional high-throughput experiment such as ChIP-seq, which must be performed in at least two replicates for each condition/sample, significantly increases the financial and labour costs of the project. However, it is important to note that for HiChIP data with poorer signal-to-noise ratio, this strategy may not be optimal and an additional ChIP-seq experiment may be required. Importantly, our pipeline also supports the integration of ChIP-seq data in such cases.

Importantly, nf-HiChIP is scalable and automated. Unlike conventional tools nf-HiChIP can process multiple samples simultaneously, making efficient use of available computational resources. The pipeline automates the processing of individual and pooled biological replicates,

and offers extensive customization options. Users can add personalised steps, select different peak calling tools, or adjust filtering processes, to meet the specific needs of their research. The integration of Nextflow[31] and Docker[62] technologies facilitates computational tasks on clusters and could be extended to analysis on cloud services such as AWS. In the future, we plan to extend the capabilities of nf-HiChIP by including more sophisticated post-processing features, such as correlation analysis between different samples and replicates, and automated generation of statistical plots. These enhancements will further streamline the workflow and provide deeper insights into the dynamics of chromatin interactions.

The visibility of architectural stripes depends on the characteristics and quality of the heatmaps obtained. In FA-EGS cohesin HiChIP the heatmap view is sharper and has less noise in comparison to cohesin FA HiChIP. Consequently, in FA-EGS HiChIP we can see finer arrangements of stripes and chains of distinct interactions in locations where standard HiChIP shows a smooth signal. These patterns can be interpreted as obstacles in the extrusion process (intermediate states of the looping process) or as a finer arrangement of loops. Similar stripes appearance is also seen in deeply sequenced Micro-C data[59,63], but not in standard Hi-C data, where stripes are more uniform[15,16]. Therefore, we believe that cohesin dcHiChIP provides more detailed dynamics of the loop extrusion process and a more complete picture of the interaction structure than the standard one. This allows the gStripe algorithm, which benefits from stronger and more numerous loops, to detect more stripes than the classical approach exemplified by the Stripenn tool, for which targets a smooth signal[30]. Overall, the greater detail of the FA-EGS heatmap data improves the distinguishability of finer stripes in the cohesin HiChIP experiment. The detection of stripes is important, as beside its relevance for loop extrusion and super-enhancer activity, they can be associated with certain pathological phenomena, such as incorrect expression of genes[64] and increased topoisomerase activity[16], which can potentially promote cancer.

We believe that our findings will be a valuable resource for researchers who encounter difficulties in their ChIP-based and high-throughput 3C-based experiments and who are looking for tools to improve and complete their bioinformatic analysis of chromatin spatial organisation data. Importantly, the use of LoopSage 3D modelling to visualise a region of

interest ensures biological realism by representing combinations of conformations that can occur in individual cells. We also demonstrate the dual interaction between polymer simulations and experimental dcHiChIP data: the loop extrusion model provides insight into the biophysical mechanisms responsible for the formation of experimentally observed structures, while the dcHiChIP data aid model validation of the computational model. Our approach can support studies that focus on the relationship between chromatin structure and the processes that occur in the nucleus such as DNA replication, transcription or DNA repair in which cohesin is an important player.

## Methods

### Cell line and culture conditions
Human HG00731 lymphoblastoid cell line, purchased from Coriell Institute for Medical Research, was grown in RPMI 1640 (Biowest) with 15% foetal bovine albumin (Biowest) and 2mM L-glutamine (Gibco). Cells were cultured at 37 °C in a humid environment containing 5% $CO_2$.

### Cell fixation
Cross-linking was performed using cells grown to a density of around 0.8 millions per 1 mL of culture medium at a volume of 1 ml of cross-linking solution per 1 million of cells. **FA cross-linking**. Pelleted cells were resuspended in a pre-warmed (37 °C) RPMI1640 (Biowest) with 1% formaldehyde (Sigma, 252549) and incubated for 10 min with agitation. Next glycine was added to a final 0.2 M concentration to quench the cross-linking reaction and the samples were incubated for 5 min with agitation at room temperature (RT). Fixed cells were then washed twice in DPBS, snap frozen in liquid nitrogen and cell pellets were stored at -80 °C. **FA-EGS cross-linking**. Pelleted cells were resuspended in a pre-warmed (37 °C) RPMI 1640 (Biowest) with 1% formaldehyde and incubated for 15 min with agitation (RT). Next glycine was added to a final 0.2 M concentration to quench the cross-linking reaction and incubated for 5 min with agitation (RT). After washing once in DPBS, cells were resuspended in pre-warmed (37 °C) DPBS with 2 mM EGS [Thermo Scientific] agitated for 45 min at RT, quenched with 0.2 M glycine and incubated with agitation for 5 min FA-EGS-fixed cells were then washed twice in DPBS, snap frozen in liquid nitrogen and cell pellets were stored at -80 °C.

### HiChIP
HiChIP assay was performed according to previously described protocol (Mumbach et al.[22]) with modifications. The combination of chromatin immunoprecipitation step with HiChIP library preparation was carried out based on ChIPmentation protocol[65] as recently reported[66]. The 150 million of crosslinked cells were resuspended in ice-cold 1.5 ml of Hi-C lysis buffer (10 mM Tris-HCl pH 8.0, 10 mM NaCl, 0.2% NP-40) supplemented with 1x protease inhibitors (PI; Roche, 04693116001), incubated on ice for 20 min and centrifuged at 1200 rpm, 5 min, 4 °C. After discarding the supernatant the pellet was resuspended in 1.5 ml Hi-C lysis buffer with 1 x PI, centrifuged at 1200 rpm, 5 min, 4 °C. Then supernatant was removed and the cell pellet was resuspended in 150ul of pre-warmed (65 °C) 0.5% SDS. After 5 min incubation at 65 °C, 435 ul of water and 75 µl of 10% Triton X-100 (Promega HP142) was added and 15 min (37 °C) incubation was carried out. Next, 75 µl of NEBuffer™ DpnII (NEB, B0543S) and 600U of DpnII (R0543M) were added to the sample and incubated for 1 h at 37 °C. To inactivate the restriction enzyme, the reaction was incubated at 65 °C for 5 min and then cooled down to RT. To perform end repair and biotinylation the following master mix was added: 67.5 µl of water, 4.5 µl of 10 mM dTTP (NEB, N0443S), 4.5 µl of 10 mM dATP (NEB, N0440S), 4.5 µl of 10 mM dGTP (NEB, N0442S), 45 µl of 1 mM biotin-16-dCTP (Jena Bioscience NU-809-BIO16-S), 24 µl of 5U/µl DNA polymerase I Large (Klenow; NEB, M0210L) and the samples were incubated for 1 h at 37 °C with rotation, Ligation was performed by addition to the sample of Ligation master mix (2007 µl of water, 300 µl of 10% Triton X-100, 18 µl of 10 mg/ml BSA (NEB, B9000S), 360 µl 10x T4 DNA ligase buffer (NEB, B0202), 15 µl of

400 U/µl T4 DNA ligase (NEB, M0202L), followed by 4 h incubation at RT with gentle rocking). Samples were then centrifuged for 5 min at 1200 rpm, at 4 °C. The nuclei pellet was resuspended in 1 ml of Nuclear Lysis buffer (50 mM Tris-HCl pH 8.0, 10 mM EDTA, 1%SDS, 1 x PI) and incubated on ice for 15 min. Sonication was carried out on Covaris S220 device in 1 ml miliTube (Covaris, 520135) using the following settings: peak power = 140, duty factor = 5, cycle burst = 200, time = 120 s, temp. = 4-8 °C. After DNA shearing, the chromatin was transferred to a new tube and 100 µl of 10% Triton-X was added. To remove cell debris, samples were centrifuged for 20 min, at 4 °C, at 16000xg. The supernatant was transferred to a new tube and diluted 5x in ChIP dilution buffer (1% Triton-X-100, 2 mM EDTA, 150 mM NaCl, 20 mM Tris-HCl pH 8.0). Next, the chromatin was pre-cleared by incubating with Dynabeads Protein A (Invitrogen, 10002D) coated with 10 µg of IgG (Abcam, AB171870) for 30 min at 4 °C, rotating. The chromatin immunoprecipitation was carried out by overnight incubation of pre-cleared chromatin with Dynabeads Protein A coated with 10 µg of SMC1 (Bethyl Laboratories, A300-055A) or 10 µg CTCF (Abclonal, A1133) antibody at 4 °C with rotation. Supernatant was then removed and the beads were washed: twice with 1 ml of Low Salt Wash Buffer (0.1% SDS, 1% Triton-X-100, 2 mM EDTA, 20 mM Tris-HCl pH 8.0, 150 mM NaCl), once with 1 ml of High Salt Wash Buffer (0.1% SDS, 1% Triton-X-100, 2 mM EDTA, 20 mM Tris-HCl p.H 8.0, 500 mM NaCl) and twice with 1 ml of 10 mM Tris pH 8.0. After careful removal of any residual 10 mM Tris pH 8.0, the tagmentation reaction was carried out by resuspending the beads in tagmentation mix (15 µl 2xTD buffer, 14 µl water), adding of 1 µl of Tn5 enzyme (TTE Mix V50, Vazyme, TD501) and incubation of the reaction for 10 min at 37 °C, with 530 rpm shaking, Next tagmentation reaction was removed and the beads were washed subsequently with 1 ml of Low Salt Wash Buffer, 1 ml of High Salt Wash Buffer, 1 ml of LiCl Wash Buffer (10 mM Tris-HCl pH 8.0, 250 mM LiCl, 1% NP-40, 1% Sodium Deoxycholate, 1 m mM EDTA) and 1 ml of 10 mM Tris pH 8.0. To elute the DNA. The beads were then resuspended in 400 ul of freshly prepared DNA Elution Buffer (50 mM NaHCO3, 1% SDS), incubated for 15 min at 65 °C. The supernatant was then transferred to a new tube and reverse crosslinking and RNA digestion was performed by adding 44 µl of mix consisting of: 20 µl 5 M NaCl, 8 µl 0.5 EDTA, 16 µl Tris 1 M pH 8.0 as well as 8 µl of 10 mg/ml RNAse (Thermo Scientific, EN0531) and incubation for 6 h at 65 °C. Proteins were then digested by 1 h incubation at 55 °C with 4 µl of 20 mg/ml Proteinase K (Ambion, AM2546). DNA was then recovered by phenol/chloroform extraction and ethanol precipitation. The DNA pellet was resuspended in 21 µl of 10 mM Tris pH 8.0, the concentration was measured with Quantus Quantifluor ONE dsDNA system (Promega, E4870) and the sample volume was increased up to 100 µl with 10 mM Tris pH 8.0. Next, 10 µl of MyOne Streptavidin C1 Beads (Invitrogen, 65001) were washed twice with 400 µl of Tween Wash Buffer (5 mM Tris-HCL pH 7.5, 0.5 mM EDTA, 1 M NaCl, 0.05% Tween-20), resuspended in 100 µl of 2x Biotin Binding Buffer (10 mM Tris-Hcl pH 7.5, 1MM EDTA, 2 M NaCl), mixed with the DNA sample and incubated for 20 min with rotation at RT. The beads were then washed (1) twice with 600 µl of Tween Wash Buffer, (2) once with 100 µl 10 mM Tris pH 8.0, (3) once with 50 µl 10 mM Tris pH 8.0 and resuspended in 23 µl of 10 mM Tris pH 8.0. To prepare sequencing libraries on-bead PCR amplification was performed with a total of 12 PCR cycles using Trueprep indexes for Illumina (Vazyme, TD202). The libraries were then sequenced on Illumina NovaSeq 6000 platform at the Genomics Core Facility, CeNT, University of Warsaw.

### nf-HiChIP pipeline
Both HiChIP and ChIP-seq datasets were processed using the nf-HiChIP pipeline. The initial stage of the pipeline mirrors a ChIP-seq analysis framework. Sequencing reads were first aligned to the hg38 reference genome employing the BWA-MEM algorithm (version v0.7.17)[67]. Subsequently, SAMtools[68] was utilised to filter the aligned reads, retaining only those with a

mapping quality (MapQ) score >30. This step was followed by deduplication of reads to eliminate potential biases arising from PCR amplification. BAM files were converted to normalised read coverage tracks (bigwig format) using the program bamCoverage in the deepTools package[43] with the binSize parameter set to default and using the reads per kilobase per million mapped reads (RPKM) normalisation option. Coverage tracks were generated using the deepTools package, facilitating the visualisation and analysis of read depth across genomic regions.

Peak calling was performed with the MACS3 software (version v3.0.1)[69] adopting a stringent q-value cutoff of 0.01 with the 'no model' option and using default value of 200 for 'extsize' parameter for the HiChIP and ChIP-seq samples to identify regions of significant enrichment. The outcomes of peak calling, applied to both ChIP-seq and HiChIP data, served as input for the MAPS algorithm[28], which delineates the final list of chromatin interactions with the anchor width of 5 kilobase (kb) from each sample. 2D contact matrix file (.hic) were calculated using Juicer tools[27] pre module from MAPS results. This analysis was conducted individually for each replicate before pooling data to create a comprehensive dataset for each sample, whereupon the entire analysis was reiterated to ensure robustness and accuracy.

Final outputs generated by the pipeline encompass a range of files for each replicate and sample, including: (i) coverage tracks, (ii) identified peaks, (iii) comprehensive MAPS output detailing the called pairwise interactions and, (iv) interaction maps compatible with the Juicebox viewer. This structured approach ensures a systematic and reproducible analysis pipeline, facilitating the in-depth exploration of chromatin dynamics and interactions. All the intermediate files (reads mapped to the reference genome, indexed BAM files, coverage tracks) can be easily accessed too.

## Juicer

All HiChIP samples were processed using the Juicer pipeline (Version 1.22.01)[27], a standard for Hi-C and HiChIP data analysis. Notably, the commonly used Knight-Ruiz normalisation (KRnorm) for Hi-C data was found inadequate for HiChIP datasets, often obscuring characteristic interactions and stripes. Consequently, VCnorm normalisation was employed with Juicer's HiCCUPS for interaction calling, circumventing the aforementioned issues. HiCCUPS uses an image processing based approach to identify loops from chromatin interaction maps, calling loops at three different resolutions: 5 kb, 10 kb, and 25 kb. This multi-resolution strategy affects the size and definition of loop anchors. Additionally, the Juicer pipeline, which includes HiCCUPS, merges nearby loops based on predefined thresholds, resulting in broader loop anchors that may encompass multiple interaction sites.

Aggregate peak analysis (APA) was conducted to assess the quality of loop calling by nf-HiChIP, HiCCUPS and ChIA-PIPE across samples. This analysis, performed via the Juicer apa function, was visualised using Matplotlib, providing a detailed comparison of loop calling efficacy. For each plot, we used the P2LL as APA score to determine enrichment of the signal to the lower left corners.

## HiChIP quality analysis and data visualisation

Post-analysis involved comprehensive sample comparisons through custom scripts in Jupyter notebook[70], utilising Numpy[71] and Pandas[72] for basic analysis, with Matplotlib and matplotlib_venn for visualisations. Sample correlation, peak enrichment, and fingerprint analysis employed deepTools[73], while genomic tracks and interaction maps were visualised using IGV[74] and Juicebox[75], respectively.

The Pearson correlation was computed for the coverage files using the multiBigwigSummary module of deeptools[73], employing default parameters to determine the average scores across various genomic regions from the provided set of bigwig files. The resulting data was then utilised by the plotCorrelation module to generate correlation coefficients, which were subsequently visualised using Matplotlib as a clustered heatmap. In this representation, the colours signify the correlation coefficients, while the clusters are formed using complete linkage.

## Loop and stripe overlap analysis

Two loops are considered overlapping if both of their anchors overlap by at least 1 base pair (bp). To allow for a 15 kb tolerance in overlap detection, the start position of one loop's anchor is extended 15 kb upstream, and its end position is extended 15 kb downstream. An overlap of at least 1 bp between the adjusted anchor of one loop and the anchor of the other loop is then checked to confirm the overlap. Similarly, two stripes are considered overlapping if (1) they have the same orientation (horizontal or vertical) and (2) their anchor regions (i.e. the 1D regions containing the coordinate where the visible stripe intersects with the diagonal) are no more than 10 kb apart. Finally, two striping domains (1D regions containing the whole extent of the stripe) are considered overlapping, if they overlap by at least 1 bp after expanding kb in both directions. These calculations are performed using custom in-house scripts and are performed for stripes, and stripe domains between all HiChIP samples.

When calculating overlaps between two sets of 1D regions (let us call them A and B), such as loop anchors, stripe anchors, domains etc., one-to-many overlaps are possible (i.e. a situation, where multiple regions from set A intersect one large region in set B, or vice versa). Therefore, to properly quantify the overlaps, we report both the number of elements from set A, for which at least one overlapping region was found in set B, and the reverse—number of elements in set B that were matched to at least one region in set A. We report this in the form of a matrix, which in the general case would not be symmetrical.

## ChIA-PIPE

The ChIA-PIPE[29] pipeline was used to process and map the CTCF HiChIP and Cohesin HiChIP data to the human hg38 reference genome. First, the reads were looked for the ligated restriction enzyme site (called a pseudo linker in HiChIP), and only the sequences that had the pseudo linker were kept for linker trimming. The flanking sequences were then mapped to the human reference genome (hg38) by Burrows Wheeler alignment (BWA), and only uniquely aligned sequences (MAPQ ≥ 30) were kept for deduplication. Next, reads with a linker sequence detected with both ends having genomic tags are used for the detection of interaction loops. These reads are categorised as self-ligation PET (both ends of the same DNA fragment) or inter-ligation PET (both ends from two different DNA fragments in the same chromatin complex). Inter-ligation PETs with a genomic span of ≥8 kb are illustrated to represent the long-range interactions of interest, further subdivided into intra- and inter-chromosomal PET clusters. Lastly, different PET clusters in the same protein factor's binding peak region were joined together with 500 bp extensions to make merged anchors. Only loops with both anchors supported by peaks identified by nf-HiChIP pipeline were retained for each sample. Only clustered and merged intra-chromosomal PETs with a PET count ≥ 3 and a genomic span < 1 MB were retained as reflecting the chromatin interactions of interest.

## hichipper

HiChIP paired-end reads were mapped to the hg38 reference genome using the HiC-Pro pipeline (version 3.1.0). Default settings were used to align paired reads to identify valid interactions and generate contact maps matrices. Then, HiChIP loops were called using hichipper (version 0.7.7) using valid HiChIP read pairs with the parameter to use pre-identified peaks by nf-HiChIP pipeline.

## FitHiChIP

Statistically significant contacts (5 kb bin size, max size 2 Mb, min size 10 kb, FDR 0.05) were identified using Hi-C Pro's allValidPairs file as input for FitHiChIP v11.0. Peaks identified by nf-hichip were used as a reference set of peaks in the FitHiChIP pipeline and default values for the remaining options.

## cLoops2

Raw paired-end reads mapped to hg38 were processed using the tracPre2.py script in the cLoops2 package. Loops were then identified with the cLoops2

callLoops module, using parameters -eps 200,500,1000, -minPts 10 and -cut 1000, requiring a minimum of 3 PETs to support each confident loop.

## CTCF motifs

To obtain CTCF motifs, we have taken a two-way approach. The first one is plainly using the motifs module from Biopython. We use it to obtain, e.g. anchors that have motifs (with a certain score - usually 7.0) in their sequence. However, to produce more detailed models, we have also introduced probabilistic motifs. In that case, we are taking all the signals of motif presence available, and we are calculating the probability of the motif being upstream or downstream-oriented using the following formula:

$$P_{right} = \frac{\sum^{right-oriented\ motifs} 2^{motif\ score}}{\sum^{all\ motifs} 2^{motif\ score}}$$

In the equation, we are taking all scores (that are in log2 form) of all motifs present in the sequence, and we convert them back from log2 form to regular one. Then, we sum the scores for the right/left orientation, and divide it by all scores from both right and left-oriented motifs. That way, we can say, e.g. that the probabilistic score of the motif being left is 30%, and right is 70%. That approach is useful for modelling and simulations.

## Annotation of loops to enhancer and promoter

To perform annotation, we downloaded cell-type-specific enhancer and promoter regions from ChromHMM chromatin state calls[76] for the GM12878 cell line from ENCODE[77]. The loop anchors were then separated into right and left anchors. Each anchor file was subsequently intersected with the enhancer and promoter files using Bedtools. Following the intersections, custom scripts were used to categorize the loops as Enhancer-Promoter (EP) or Promoter-Promoter (PP) interactions.

## Calling architectural stripes

Stripe calling was performed using two tools: the Stripenn[30], which operates on heatmap data, and gStripe, which was developed for calling stripes from sets of discrete interactions, such as loops obtained from other tools. The Stripenn is based on image analysis and computer vision methods. It processes the contact matrix obtained from a 3 C experiment using a series of computer vision techniques, most importantly the canny edge detection, to select regions with high "stripeness" scores, i.e. those that are visually consistent with a horizontal or vertical stripe. It also calculates the p-value of a stripe, based on the distribution of the values taken from the same heatmap, and this provides the final significant stripe set. To ensure our analyses reflect the best possible extraction of useful information from each dataset, we tuned two of Stripenn's parameters (the Canny edge detection parameter and the kernel size of the mean filter) to obtain the maximum number of stripes in each dataset. This was done by a simple grid search centred around the values suggested by the authors.

As an alternative to image processing methods, we also include the results from our own tool, gStripe, designed specifically for sparse data. In contrast to contact matrix-based methods, gStripe operates on sets of loops obtained from other tools, in our case nf-HiChIP, HiCCUPS and ChIA-PIPE. The algorithm can be divided into three parts: preprocessing of input data, identification of candidate stripes, and selection of final stripes. During preprocessing all input loop anchor coordinates are binned at 5 kb resolution, to avoid any bias introduced by comparison between binned and non-binned input data. Next, a graph representation of the chromatin interactions is created. Continuous clusters of overlapping anchors are merged to represent a single vertex in the graph. The edges are then added to the graph, where there are interactions between anchors assigned to two different vertices. Multiple edges connecting the same vertex are collapsed into a single edge, with weight equal to the number of interactions (hence, the resulting graph is not a multigraph). The physical position of each vertex is calculated as the centre of the anchor cluster forming the vertex.

To identify the candidate stripes, each vertex with at least one downstream neighbour in the graph is selected as a potential anchor of a horizontal stripe, i.e. a stripe which appears as a horizontal flare above the diagonal on a heatmap, or whose anchor (the location where the stripe reaches the diagonal) precedes its other end in terms of genomic coordinates. Likewise, each vertex with at least one upstream neighbour is selected as a potential vertical stripe (appearing as vertical pattern above diagonal on a heatmap, with the anchor being downstream of the stripes longest extent). We will now refer to these anchor-forming vertices simply as "anchors", and to the other vertices in a candidate stripe as "leaves". A number of values is computed for each leaf $v_i$, in order to measure the quality of the stripe. Let $v_i = 1, 2, ..., k$ denote the k leaves of a stripe, and $v_0$ denote the anchor.

1. The length of the stripe at the leaf $v_i$: $d_i = |v_i - v_i|$. Prior to this step, the vertices are arranged so that $d_i$ is ascending with i, i.e. $v_0$ is the anchor, $v_1$ is the vertex closest in terms of genomic coordinates to the anchor, etc. In a horizontal stripe, the coordinates of $v_i$ have increasing order, and in a vertical stripe they have a decreasing order.
2. Relative gap $g_i = -\log \frac{d_i - d_{i-1}}{d_i}$. All lengths are set to be at least 1 kb to ensure that this value is well-defined.
3. $q\_gap_i$ - quantile of the relative gap values across all candidate stripes in the dataset identified in the previous step.
4. $stripe\_score_i$ - quadratic mean of the $q\_gap_j$ for j = 1,…, i. Represents the average quality of the stripe of up to a certain point (i.e. leaf $v_i$).
5. $cross\_score_i$ - each leaf is potentially a part of two orthogonal stripes: horizontal and vertical one. For a given stripe direction the cross score of $v_i$ is the ratio of $stripe\_score_i$ calculated in that direction and the stripe score in the orthogonal direction.
6. The length of the stripes are determined: a stripe is considered to end with a leaf $v_i$, if $g_{i+1}$ drops below 0.05, or if there are no further leaves.
7. If two adjacent stripes with the same directionality overlap and differ in length by no more than 200 kb, they are merged together.
8. Only stripes with at least two leafs, and having a minimum length of 20 kb are selected and considered further.

The above steps result in a set of candidate stripes. In order to select the final stripes from the candidate set, two features of the stripes are considered: the $stripe\_score$ of the leaf terminating the stripe (note that this is a form of cumulative quality of the stripe up to this leaf), and the mean $cross\_score$ of every leaf in the stripe after its length has been trimmed. These parameters measure the continuity and distinctiveness of the stripe respectively, and the selection of the final stripe set is done by placing a lower threshold on these values. The choice of the thresholds is done by visual inspection of the output and manual adjustment; this does not require, however, re-running the algorithm. We used the thresholds of 0.45 for $stripe\_score$ and 0.9 for $cross\_score$. Note that in contrast to methods based on heatmap analysis (such as Stripenn), potential stripe locations are already narrowed down, as the stripe can only be called in a location where two consecutive loops were present.

Pileup plots for the stripe analysis were constructed using Coolpuppy[78] API. In line with the methodology used in Stripenn[30] we used parameters equivalent to "--rescale --local --unbalanced" options. In this procedure each stripe domain is expanded by regions of equal size from both sides, then rescaled to a standard size, and the raw contact matrix signal is averaged over all stripe domains in a given dataset.

## Biophysical modelling

For loop extrusion modelling, we used LoopSage[32]. We assume that the stochastic system can be described by a Boltzmann distribution, where the temperature parametrises the rebinding probability of cohesin[79]. During the simulation, the temperature decreases following a simulated annealing approach[80]. The dcHiChIP dataset inspired us to develop a new, more user-friendly environment for LoopSage. This environment includes the capability to model two different populations of loop extruders that move at different speeds and diffuse as random walks. This observation aligns with previous work from Gibcus et al.[38], which proposed two populations of extruders capable of forming short and long-range loops.

**Article**

The simulation pipeline is composed of two main parts: a stochastic simulation part, where we compute the stochastic trajectories of cohesins in the DNA fibre by running the Monte Carlo simulation, and a molecular simulation part, where we import the cohesin positions exported by the numerical simulation part and we feed them into an OpenMM model[81–83] which is capable of producing 3D-structures from them. Having produced the ensemble of models, we compute the average inverse all-versus-all distance heatmap of each one of these 3D models. Finally, we aggregate them into an average final heatmap, which should reconstruct similar patterns as the experimental HiChIP heatmap. For the purposes of this work, we used a polymer that consists of 1000 monomers. The stochastic simulation ran for $8 \times 10^4$ Monte Carlo steps with sampling frequency 400 and burn-in period $4 \times 10^4$. The value of the folding and crossing coefficients are set as the default ones. The reason why we chose such a large burn-in period is because we wanted to have enough time to propagate the slow cohesin complexes. We applied a simulated annealing approach with initial Monte Carlo temperature 4 and final temperature 1, assuming that representative steps emerge once slower, shorter loops begin to form after the burin-in period.

To validate our models, we employed a two-step approach. First, we estimated the Pearson correlation between experimental and averaged simulated heatmaps, focusing only on regions where loops were detected in the experimental data (loop-strength correlation). This initial validation was straightforward because LoopSage uses anchors and their averaged strengths as input, resulting in consistently high correlations. In the second validation step, we compared the entire simulated heatmap with the experimental one. To facilitate this comparison, we applied Gaussian smoothing to the experimental heatmap and calculated Spearman correlations for various LoopSage parameters.

We made a qualitative study about the variation of the resulting averaged inverse distance heatmaps based on two parameters of interest: (i) the number of cohesin molecules (tested for values 50, 100, 150), and (ii) the binding coefficient of the CTCF motif as it is defined in our new documentation (tested for values 0.25, 0.5, 1). Strong CTCF binding with high cohesin levels creates overly confined loops, whereas weak binding with few cohesins yields broad TADs lacking distinct borders (Supplementary Fig. 12A). To find the best choice of model parameters we chose the model with the highest Spearman correlation between simulated and experimental heatmaps (Supplementary Fig. 12B). To achieve higher correlation, it is required adjusting excluded volume parameters and introducing a second, faster family of extruders. With 100 slow and 5 fast extruders, plus an excluded volume power of 3, we matched better both short- and long-range loops. While changing excluded volume had limited effects, adding a few fast diffusing extruding factors enabled proper long-range loop formation, with slower extruders then forming local loops as seen in the 3D models and averaged heatmaps (Supplementary Fig. 12B).

## Statistics and reproducibility

For SMC1 and CTCF HiChIP (HG00731) experimental data, two independent biological replicates were performed. Information on replicates and reproducibility is provided in the Results section and Supplementary Figs. 1-2.

## Reporting summary

Further information on research design is available in the Nature Portfolio Reporting Summary linked to this article.

## Data availability

Sequencing data generated in this study is deposited in the Gene Expression Omnibus (GEO) database, with an accession number GSE266640. Raw data and analysed data that support the findings of this study are also submitted 4D Nucleome data repository[84,85] (https://data.4dnucleome.org) under accession number 4DNES7BZ38JT and 4DNESOAF3QAA.

## Code availability

nf-HiChIP pipeline is available at https://github.com/SFGLab/hichip-nf-pipeline. The pipeline is implemented in Nextflow with Docker support and processes the output of various tools. The gStripe algorithm implementation is available at https://github.com/SFGLab/gStripe as a Python 3 package. The latest version of LoopSage is available via PyPI https://pypi.org/project/pyLoopSage/ and as an open-source project on GitHub at https://github.com/SFGLab/pyLoopSage. The source code for all the pipelines can also be found at https://doi.org/10.5281/zenodo.11213538.

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

## Acknowledgements
We would like to thank Victor Corces and the Corces lab for the introduction to the HiChIP assay and analysis, Dorota Adamska from Genomics Core Facility (CeNT, UW) for designing the sequencing strategy and providing constant support in trouble shooting experiments, our laboratory members for discussions and Veronika Mančíková for proofreading of the manuscript. This project received funding from the Polish National Science Centre (2019/35/O/ST6/02484 to D.P., 2020/37/B/NZ2/03757 to D.P., 2020/04/X/NZ2/01006 to K.B.), Boehringer Ingelheim Fonds travel grant to K.B., Warsaw University of Technology within the Excellence Initiative: Research University (IDUB) programme, National Institute of Health USA 4DNucleome grant 1U54DK107967-01 "Nucleome Positioning System for Spatio-temporal Genome Organization and Regulation", EU-funded Innovative Training Network "Molecular Basis of Human Enhanceropathies" (Enhpathy, www.enhpathy.eu), the Horizon 2020 research and innovation programme under the Marie Sklodowska-Curie grant agreement No 860002 and Excellence Initiative: Research University (IDUB) programme with

Agreement no. BOB-IDUB/IV.4.1/4/2024 to AA, Polish National Agency for Academic Exchange (PPN/STA/2021/1/00087/DEC/1 to MC). Computations were performed thanks to the Laboratory of Bioinformatics and Computational Genomics, Faculty of Mathematics and Information Science, Warsaw University of Technology using Artificial Intelligence HPC platform financed by Polish Ministry of Science and Higher Education (decision no. 7054/IA/SP/2020 of 2020-08-28). NGS was performed thanks to Genomics Core Facility CeNT UW (RRID:SCR_022718), using NovaSeq 6000 platform financed by Polish Ministry of Science and Higher Education (decision no. 6817/IA/SP/2018 of 2018-04-10).

## Author contributions
K.B., A.A., Z.P.-T., and D.P. conceived the study. K.B. performed HiChIP experiments; Data analysis and pipeline development was performed by Z.P.-T. and A.A. with assistance from M.C., K.H.B and D.P. M.D. performed stripe calling and analysis, and S.K. carried out loop extrusion modelling. D.P. and K.B. designed and supervised the study. K.B., A.A., and Z.P.-T prepared the manuscript with assistance from M.C., M.D., S.K. and D.P. All authors discussed the results and commented on the manuscript.

## Competing interests
The authors declare no competing interests.
