## [Transparent Peer Review file · Communications Biology]

Improved cohesin HiChIP protocol and bioinformatic analysis for robust detection of chromatin loops and stripes

Corresponding Author: Professor Dariusz Plewczyński

Version 0:

Reviewer comments:

Reviewer #1

(Remarks to the Author)

Jodkowska et al are reporting an improved cohesin HiChIP protocol (Hi-C with pull down of cohesin mediated interactions). Using additional crosslinking agent EGS on top of typically used formaldehyde, authors achieve improved signal-to-noise ratio. This improvement is quantified using multiple metrics: (1) enrichment of cohesin peaks derived from HiChIP coverage tracks ("1D peaks") and comparison of these peaks with related cohesin ChIP-seq and CTCF HiChIP/ChIP-seq (2) enrichment of cohesin-mediated chromatin "loops" detected using 3 different methods ("2D peaks"), etc.

Signal-to-noise improvements reported in this study are in line (and expected) with the prior studies that used additional crosslinkers (EGS and/or DSG) to improve signal-to-noise of both Hi-C and ChIP-seq protocols - these studies are properly cited and acknowledged by the authors.

In addition to the experimental protocol authors present nextflow-based HiChIP/ChIP-seq data processing pipeline "nf-HiChIP". This pipeline "glues" together established bioinformatics tools (bwa, deeptools, MACS for ChIP-seq, MAPS for HiChIP/PLAC-seq, etc) and automates processing of multiples samples and replicates. The pipeline "deliverables" include both 1D (protein binding peaks, HiChIP/ChIP coverage tracks - "deliverables" of a regular ChIP-seq) and 2D results: significantly enriched pairwise interactions.

Manuscript is written in great detail and reported findings are supported by the analyses of the data.

I do have a few minor comments/suggestions:

1. Loop-extrusion based "TAD"-domain is not a universal feature of DNA organization over the entire evolutionary tree (not even eukaryotes e.g. Matthews, N.E. and White, R. (2019), Chromatin Architecture in the Fly: Living without CTCF/Cohesin Loop Extrusion?. BioEssays, 41: 1900048. <https://doi.org/10.1002/bies.201900048>) - in the 2nd paragraph of "Introduction" - please narrow down the statement to relevant evolutionary class/subclass - mammals, vertebrates ?
2. Moreover, direct observation of loop extrusion pioneered in the works of Cees Dekker and others arguably allows to forgo historic term "TAD" and talk about loop extrusion as the main driver of chromatin organization (at least in mammals) - where "TAD"-domains, "loops" and "stripes" are just the population average manifestations of the extrusion as detected by 3C-based methods
3. In the second section of "Results", 3rd paragraph, authors state that one can expect more cohesin peaks to be detected from the HiChIP coverage track (vs regular ChIP-seq) - because HiChIP peaks may correspond to regions that are bound by target protein or regions that are in contact with the protein binding site - (1) should the first part refer to DNA in proximity of the target protein ? as being bound sounds similar to being in contact with the binding site ? (2) even still - isn't this statement relevant for regular ChIP-seq as well - i.e. DNA undergoes fixation in ChIP-seq protocol, thus binding DNA directly bound by proteins and DNA that is near a protein (~16 angstrom for EGS) (3) are you referring to the ligation related biases that can be introduced in HiChIP, as the proximity ligation is the step that is missing from a regular ChIP-seq ?
4. I was surprised to find out that "nf-HiChIP" does not yield an interaction heatmap as an output, only significant interactions detected by MAPS. I think it would be useful to add such a functionality in a development roadmap or at least refer user to existing HiC pipelines that can deliver that (mentioning HiChIP specific needs for normalizations/balancing or lack of it)
5. Loop extrusion simulation part feels detached from the rest of the manuscript - could authors provide more reasoning behind including it ? Could authors also clarify if simulated heatmap represent *all* chromatin interactions in the region (i.e. simulated HiC) or does it try to simulate "cohesin"-mediated interactions (i.e. HiChIP-itself) - would that help explaining some

apparent misalignment of extrusion features on Fig 3E (stripe at the position ~180, small domain around ~110-125, lack of the distant "loop" connecting ~25 and 180 in the simulations) ?

Reviewer #2

(Remarks to the Author)

The manuscript from Jodkowska and coworkers describes an improved HiChIP protocol that employs dual crosslinkers to enhance the detection of chromatin interactions. Additionally, the manuscript presents an automated analysis workflow called nf-HiChIP and a stripe-calling method named gStripe. While the approaches could potentially be interesting, several methodological concerns have to be addressed to show the practical value as outlined below.

1. The cohesin ChIP-seq data used in the study requires further clarification. The supplementary table2 presents SMC1 ChIP-seq data, while the manuscript states that the ChIP-seq data is derived from the SA1 subunit. If data for SMC1 is unavailable and SA1 is selected, why were the data for SMC3 or Rad21, which are also core cohesin proteins and have binding sites more consistent with SMC1, not chosen?
2. The authors state in the second paragraph on page 13 that "ChIP-seq experiments for HG00731 were not available". Theoretically, since the HiChIP process involves immunoprecipitating the target protein after the standard Hi-C ligation step, if a HiChIP experiment can be performed, the corresponding ChIP-seq experiment should be relatively easy to conduct.
3. It was noted that over 600,000 peaks were obtained from the cohesin HiChIP data, many of which do not show a clear distinction from the background. The source article for this data (Mumbach, 2016) identified approximately 28,000 peaks, whether the parameters chosen for peak calling were appropriate?
4. In the nf-HiChIP pipeline, HiChIP data was used instead of ChIP-seq data to obtain signal peaks and subsequently annotate loops. Testing with public data revealed that high signal-to-noise ratio data is necessary. For data with a moderate signal-to-noise ratio (e.g., GSM2774002, HiChIP_K562_YY1), the pipeline reported errors and stopped running. Additionally, it was noted that the pipeline does not support stepwise execution; if an error occurs at a specific step or if the code is interrupted, it cannot resume from that step and must be restarted from the beginning.
5. In the second paragraph on page 16, line 12, the authors state that approximately 75% to 90% of the nf-HiChIP loops overlap with those detected by HiCCUPS. Given that the manuscript mentions the two methods detected 80,487 and 43,148 loops, respectively, this ratio raises concerns.
6. The authors detected loops from cohesin FA-EGS HiChIP data using three different methods: nf-HiChIP, HiCCUPS, and ChIA-PET. If additional data, such as CTCF binding patterns, associations with promoters or enhancers, and types of histone modifications, could be incorporated, it would provide more compelling evidence for evaluating the reliability of loops annotated by different methods based on potential biological relevance.
7. Since the gStripe algorithm relies on the detected loops, it would be beneficial to provide an evaluation or application standard instead of merely reporting detection rates among different methods. This would help determine the suitability of the loops obtained using various methods, particularly nf-HiChIP and ChIA-PET, for analyzing the intended data.
8. The description of data overlap in the manuscript is not clear enough, for example, figure 2D requires consulting the figure legend for proper understanding.
9. A token or temporary link to access the data was not provided by the authors.

Minor Issues

1. there are some issues with the supplementary tables in the PDF version, supplementary table1 is split, and tables2-7 are missing.
2. The order of some supplementary figures in the manuscript does not follow the numbering, for example, Supplementary Figure1B appears before Supplementary Figure1A, and Supplementary Figure6 is not mentioned in the manuscript.
3. In line 9 of the second paragraph on page 2, it should be "10-nm chromatin fiber".
4. In line 10, it seems should be "a subset of loop anchors has been shown to act as boundaries of TADs."
5. In line2 of the second paragraph on page 3, there is a missing closing parenthesis after "reviewed in 24".
6. In the HiChIP experimental method, line 9 on page 5, "resuspending in 150 µl of pre-warmed" seems to be missing SDS and its concentration.
7. In the second-to-last line of the second paragraph on page 13, it should be "the latter type of experiment."
8. In the first line of the last paragraph on page 14, it should be "to verify the influence of dual cross-linking on HiChIP."
9. In the second paragraph on page 17, line 6, "ChIP-PIPE" should be "ChIA-PIPE".

Reviewer #3

(Remarks to the Author)

The work by Jodkowska, Parteka-Tojek, and Agarwal introduces an improved HiChIP protocol with dual cross-linking for cohesin, which enhances the signal-to-noise ratio and includes the analysis pipeline nf-HiChIP. They demonstrate a comprehensive improvement in HiChIP signals at both 1D peaks and 2D loops levels and develop biophysical models of loop extrusion using the data, which is intriguing. However, the concept of dual cross-linking lacks novelty, and the pipeline primarily integrates existing tools without introducing significant algorithmic innovation, which may limit the impact of the work. In my view, the manuscript could benefit from additional analyses, as outlined in the following major suggestions:

Major suggestions:

- 1) It would be beneficial to test if the dual cross-linking method is also effective for other low-affinity factors such as YY1. Expanding the method to include factors like Wapl could significantly broaden the method's applicability and deepen the biological insights obtainable.
- 2) The results section contains redundant descriptions in the first and fourth paragraphs concerning the focus on human lymphoblastoid cell lines (LCLs) from the 1000 Genome project and the assessment of the dual cross-linking approach's quality. Streamlining these descriptions would enhance clarity and reader comprehension.
- 3) To detect loops connecting two distant genomic fragments, the study employs three independent algorithms. I recommend including additional algorithms such as hichipper (Lareau and Aryee, 2018) and cLoops2 (Cao et al., 2022) to potentially improve loop detection accuracy. It would also be informative to include the number of loops called for different datasets and by different software in Figures 2C, S4C, and S5C.
- 4) Supplementary Figure 7A is mentioned before Supplementary Figure 6 in the text, and there appears to be a reference missing to Supplementary Figure 6. Adjusting the order and ensuring all figures are correctly referenced would improve the logical flow.
- 5) I suggest creating a figure similar to Figure 11 using the cLoops2 estRes to showcase interaction resolution improvements, with data downsampled to the same final read depth for a consistent comparison.

These improvements could significantly strengthen the manuscript by broadening its methodological scope and enhancing the clarity and depth of the presented data and analyses.

Version 1:

Reviewer comments:

Reviewer #1

(Remarks to the Author)

Authors have addressed all of my concerns and comments, and the manuscript appears to be significantly improved.

Minor:

There is a typo in the title of one of the methods section: "annotation of loops to enhance(R) and promoter"

Reviewer #2

(Remarks to the Author)

Thanks to the authors for providing detailed responses to my previous comments and suggestions. After reviewing the revised manuscript, I find that it has significantly improved in both readability and content compared to the initial version. The presentation of the results is clearer and more comprehensive. The authors have addressed my concerns thoroughly and made appropriate revisions to the manuscript, and at this point, I have no further questions or suggestions.

Reviewer #3

(Remarks to the Author)

Most of my comments have been well addressed. I still want to see the cLoops2 estRes results of the HiChIP data as a file reply to reviewer only. Thanks.

Reviewer 1:

1. Loop-extrusion based “TAD”-domain is not a universal feature of DNA organization over the entire evolutionary tree (not even eukaryotes e.g. Matthews, N.E. and White, R. (2019), Chromatin Architecture in the Fly: Living without CTCF/Cohesin Loop Extrusion?. BioEssays, 41: 1900048. <https://doi.org/10.1002/bies.201900048>) - in the 2nd paragraph of “Introduction” - please narrow down the statement to relevant evolutionary class/subclass - mammals, vertebrates ?

R1: We agree with the reviewer that the partitioning of the genome into TADs is not a general feature of eukaryotes. We are grateful for the reference to the relevant publication that addresses this issue. As suggested, we have restricted the description of the levels of chromatin organisation in the nucleus to mammals (introduction, 2nd paragraph, 1st sentence).

2. Moreover, direct observation of loop extrusion pioneered in the works of Cees Dekker and others arguably allows to forgo historic term “TAD” and talk about loop extrusion as the main driver of chromatin organization (at least in mammals) - where “TAD”-domains, “loops” and “stripes” are just the population average manifestations of the extrusion as detected by 3C-based methods.

R2: We agree that loop extrusion is a key process that produces loops, most TADs, and stripes. We recognize that our introduction to this phenomenon may have been somewhat misleading to the reader (e.g., presenting TADs and loops as different levels of chromatin organization). However, since (1) not all TADs are flanked by CTCF binding sites (Rowley and Corces, 2018); (2) not all domains are demarcated by loops (Rao et al., 2024) and (3) TAD is still a widely used term in the field (see for example the recent review by Zhang et al. (2024), we have decided to retain the term TAD/domain as a feature of the genome.

We have incorporated the following changes to the text to make the introduction to this topic clearer and more precise:

- Introduction, paragraph 2

Replace:

“Furthermore, Topologically Associated Domains (TADs) can be detected within the compartments. TADs are sub-megabase regions that have stronger contacts within themselves than with other regions 4–6. At a finer scale, the 10 chromatin fibre folds into loops which connect elements that may be distant in the linear genome (such as promoters and enhancers).”

With the following sentence:

“Within the compartments, the 10-nm chromatin fibre folds into loops connecting elements that may be distant in the linear genome, such as promoters and enhancers. A subset of chromatin loops form larger structures called Topologically Associated Domains (TADs) or domains, which are sub-megabase regions that have stronger contacts within themselves than with other regions 4–6.”

- Introduction, paragraph 3 -

Remove the words "TADs and" from the first sentence.

- Introduction, paragraph 4 -

Remove the words "TADs and" from the first sentence.

Delete the last sentence: "While CTCF-mediated interactions are conserved to a great extent across cell types and during differentiation, CTCF-independent interactions have a more dynamic nature; they are cell type-specific, associated with gene expression, and often reside at enhancers 22,23", as it is not critical to the understanding of the following sections of the manuscript.

- Introduction, paragraph 6

Replace the 2nd sentence:

"Stripes are postulated to be the effect of loop extrusion, where one loop anchor is held in place, and becomes the stripe anchor, and the other loop anchor moves through the stripe domain until the loop is fully extruded 28,29"

With the following sentence:

"Similar to loops and TADS, stripes are the population average representation of loop extrusion as detected by the 3C-based methods, where one loop (stripe) anchor is held in place, and the other loop anchor moves through the stripe domain until the loop is fully extruded 28,29."

- Introduction, paragraph 6

Move the last sentence of paragraph 6: "During the extrusion process, promoters can be brought into contact with long clusters of enhancers 17", to the end of paragraph 4, which describes the loop extrusion process.

References -

- Rowley, M. J., & Corces, V. G. (2018). Organizational principles of 3D genome architecture. *Nature reviews. Genetics*, 19(12), 789–800. <https://doi.org/10.1038/s41576-018-0060-8>
- Rao, S. S., Huntley, M. H., Durand, N. C., Stamenova, E. K., Bochkov, I. D., Robinson, J. T., Sanborn, A. L., Machol, I., Omer, A. D., Lander, E. S., & Aiden, E. L. (2014). A 3D map of the human genome at kilobase resolution reveals principles of chromatin looping. *Cell*, 159(7), 1665–1680. <https://doi.org/10.1016/j.cell.2014.11.021>
- Zhang, Y., Boninsegna, L., Yang, M., Misteli, T., Alber, F., & Ma, J. (2024). Computational methods for analysing multiscale 3D genome organization. *Nature reviews. Genetics*, 25(2), 123–141. <https://doi.org/10.1038/s41576-023-00638-1>

3. In the second section of "Results", 3rd paragraph, authors state that one can expect more cohesin peaks to be detected from the HiChIP coverage track (vs regular ChIP-seq) - because HiChIP peaks may correspond to regions that are bound by target protein or regions that are in contact with the protein binding site - (1) should the first part refer to DNA in proximity of the target protein ? as being bound sounds similar to being in contact with the binding site ? (2) even still - isn't this statement relevant for regular ChIP-seq as well - i.e. DNA undergoes fixation in ChIP-seq protocol, thus binding DNA directly bound by proteins and DNA that is near a protein (~16 angstrom for EGS) (3) are you referring to the ligation related biases that can be introduced in HiChIP, as the proximity ligation is the step that is missing from a regular ChIP-seq ?

R3: We are grateful to the reviewer for pointing out this inaccuracy. We agree that this statement is also relevant for regular ChIP-seq experiments. Therefore, differences between the results of ChIP-seq and HiChIP experiments performed for the same protein factor should be attributed to the chromatin conformation capture step, which is not present in a ChIP-seq experiment.

- For that reason, we decided to remove the 2nd and 3rd sentences from the mentioned paragraph:

“Importantly, in the HiChIP experiment, peaks may correspond to regions that are bound by a target protein or to regions that are in contact with the target protein binding sites. Therefore, a higher number of peaks can be expected for the HiChIP experiment compared to the ChIP-seq experiment.”

- To explain the discrepancy between results from both experiments, we added the following sentence in the middle of the paragraph:

“We attribute the difference in peak counts between the HiChIP and ChIP-seq experiments to the biases of the 3C step, which is not present in the regular ChIP-seq experiment.”

4. I was surprised to find out that “nf-HiChIP” does not yield an interaction heatmap as an output, only significant interactions detected by MAPS. I think it would be useful to add such a functionality in a development roadmap or at least refer user to existing HiC pipelines that can deliver that (mentioning HiChIP specific needs for normalizations/balancing or lack of it).

R4: We would like to thank the reviewer for his valuable suggestion. Based on the feedback, we have updated the nf-HiChIP pipeline (<https://github.com/SFGLab/nf-hichip>), which now generates a .hic heatmap as an output that is compatible with the Juicebox viewer. We have made the following updates in the ‘Materials and methods’ section to incorporate this information.

- In “Data Processing and Analysis” section, and subsection “nf-HiChIP pipeline”, paragraph 2 - *2D contact matrix file (.hic) were calculated using Juicer tools 30 pre module from MAPS results.*
- In “Data Processing and Analysis” section, and subsection “nf-HiChIP pipeline” paragraph 3 - *and, (iv) interaction maps compatible with the Juicebox viewer.*
- We have also updated Figure 4 to align the updated function of the pipeline

5. Loop extrusion simulation part feels detached from the rest of the manuscript - could authors provide more reasoning behind including it? Could authors also clarify if simulated heatmap represent *all* chromatin interactions in the region (i.e. simulated HiC) or does it try to simulate “cohesin”-mediated interactions (i.e. HiChIP-itself) - would that help explaining some apparent misalignment of extrusion features on Fig 3E (stripe at the position ~180, small domain around ~110-125, lack of the distant “loop” connecting ~25 and 180 in the simulations)?

R5. We thank the reviewer for the thoughtful feedback on the loop extrusion simulation section of our manuscript. We understand that this part may have seemed disconnected from the rest of the study, which focuses on experimental comparisons with publicly available HiChIP data. The primary purpose of including the loop extrusion simulations is to provide a deeper insight into the underlying biological processes that govern chromatin organisation, in particular the dynamics of cohesin. By integrating

Figurebiophysical modelling with our experimental data, we aim to bridge the gap between observed chromatin interactions and the mechanisms that drive them. This approach allows us not only to validate our experimental findings, but also to improve our understanding of the role of cohesin in shaping chromatin architecture. In addition, the experimental data can help us to improve the simulation process. We have added additional explanation regarding the integration of dcHiChIP data with loop extrusion simulation in the subsection "Dual cross-linking HiChIP protocol allows us to build accurate loop extrusion models" of the Results section as well as in the last paragraph of the Discussion section.

The simulated heatmaps represent cohesin-mediated interactions, analogous to HiChIP data, rather than all chromatin interactions captured by Hi-C. Thus, LoopSage focuses primarily on the extrusion process and cohesin dynamics, while also incorporating the positions and strengths of CTCF motifs as barriers to cohesin movement. By taking into account CTCF binding sites, the model simulates how CTCF influences the ability of cohesin complexes to extrude loops, which is critical for accurately reproducing the patterns observed in experimental data. By focusing on cohesin-mediated interactions, we can directly compare our simulations with experimental HiChIP results and assess the effectiveness of our model in reproducing observed patterns.

We are grateful for pointing out the misalignment (stripe at position ~180, the small domain around ~110-125). It was due to technical error while aligning the simulated heatmap with the experimental heatmap created by the LoopSage.

To address the lack of the distant "loop" connecting ~25 and 180 we have refined our model. First we have made a qualitative study about the variation of the resulting averaged inverse distance heatmaps based on two parameters of interest: (i) the number of cohesin molecules (tested for values 50, 100, 150), and (ii) the binding coefficient of the CTCF motif as it is defined in our new documentation (tested for values 0.25, 0.5, 1). Strong CTCF binding with high cohesin levels creates overly confined loops, whereas weak binding with few cohesins yields broad TADs lacking distinct borders (Supplementary Figure 1A). To find the best choice of model parameters we chose the model with the highest Spearman correlation between simulated and experimental heatmaps (Supplementary Figure 1B, Left). Second we introduced two different classes of cohesin with different loop extrusion speeds. This modification is inspired by a study from the lab of Job Dekker (Gibcus et al. 2018) suggesting that extruding factors can exhibit variable dynamics. Specifically, we added to our model a second population of faster-moving extruding factors (cohesin complexes) in addition to the slower-moving ones initially modeled. The fast-moving cohesin complexes are fewer in number but can form long-range loops more rapidly, capturing distant interactions observed in the experimental data.

These adjustments have enhanced the alignment from 98.3% to 98.4% for simulated loop strength heatmaps and from 74% to 81% for entire heatmaps (Supplementary Figure 1). The long-range loop between ~25 and 180 has improved and is now better visible in the simulated heatmap in the updated Figure 3E. Some discrepancies between the simulated and experimental heatmap can be attributed to the current limitations of the model and its stochastic nature. The changes to the modelling process have resulted in updates to Figures 3D and 3E, the addition of a new figure (Supplementary Figure 1), and expanded explanations in the 'Biophysical Modelling' subsection of the 'Materials and Methods' section and the 'Dual Cross-Linking HiChIP Protocol Allows Accurate Loop Extrusion Models' subsection of the 'Results' section.

References -

- Gibcus, J. H., Samejima, K., Goloborodko, A., Samejima, I., Naumova, N., Nuebler, J., Kanemaki, M. T., Xie, L., Paulson, J. R., Earnshaw, W. C., Mirny, L. A., & Dekker, J. (2018). A pathway for mitotic chromosome formation. *Science (New York, N.Y.)*, 359(6376), eaao6135. <https://doi.org/10.1126/science.aao6135>

Reviewer 2:

Major Comments -

1. The cohesin ChIP-seq data used in the study requires further clarification. The supplementary table2 presents SMC1 ChIP-seq data, while the manuscript states that the ChIP-seq data is derived from the SA1 subunit. If data for SMC1 is unavailable and SA1 is selected, why were the data for SMC3 or Rad21, which are also core cohesin proteins and have binding sites more consistent with SMC1, not chosen?

R1. We thank the reviewer for the insightful comment regarding the cohesin ChIP-seq data used in our study. We apologise for any confusion caused by the discrepancy between Supplementary Table 2 and the manuscript. We are grateful for the opportunity to clarify our data selection process.

- We mistakenly wrote SMC1 instead of SA1. Therefore, the following change has been made in Supplementary Table 2 as well as in the Supplementary Table 1.

From SMC1 GM1278 to SA1 GM12878

- Moreover, for better clarity and identification of the datasets, in Supplementary Table 1 we have added information about the Accession Id for each dataset i,e HiChIP and ChIP-seq used in the manuscript.

Our initial goal was to utilise the same SMC3 GM12878 ChIP-seq dataset (Encode ID : ENCSR000DZP) reported in the Mumbach et al. (2016) publication to maintain consistency with previous studies. However, upon further investigation, we identified newer ChIP-seq experiments released by the Snyder Lab, which originally published the SMC3 dataset. These newer datasets included experiments for the cohesin SA1 subunit (Accession: GSM1233911), CTCF (Accession: GSM1233887, GSM1233888) and corresponding input controls (Accession: GSM1233908). These datasets were sequenced at higher depth and utilised a paired-end sequencing approach, unlike the single-end sequencing used in the original SMC3 dataset, resulting in superior data quality.

Upon processing the SA1 ChIP-seq sample, we compared it with the previously mentioned SMC3 ChIP-seq dataset. Our analysis demonstrated that the SA1 ChIP-seq data recovered over 90% (31,427 out of 34,390) of the original SMC3 peaks, while also identifying an additional 28,742 binding sites (Figure R1, upper panel). This suggests that the SA1 dataset is more comprehensive, as it not only captures the majority of SMC3 binding sites but also detects additional cohesin binding regions. Another crucial factor in our decision was the availability of corresponding CTCF ChIP-seq and input samples from the same project which enhanced the reliability of our comparative analyses and minimised potential batch effects.

- We have therefore added the following sentence to the first paragraph of the Results section:

“We used SA1 ChIP-seq dataset⁶⁰ for comparative analysis because of two reasons: (1) SA1 ChIP-seq data recovered over 90% of the SMC3 ChIP-seq peaks⁴⁹ used for the cohesin HiChIP analysis by Mumbach et al., 2016, while also identifying nearly 30000 additional binding sites (Supplementary Figure 2B), and (2) the corresponding CTCF ChIP-seq was available for the same experimental setup enhancing the reliability of our comparative analyses by reducing potential batch effects.”

- Additional figure (Supplementary Figure 2B) has been added to show the Overlap of peaks

from different GM12878 cohesin ChIP-seq datasets. - GM12878 SA1 ChIP-seq and GM12878 SMC3 ChIP-seq.

Moreover, at the time of our study, we were unable to identify cohesin ChIP-seq datasets with comparable sequencing depth and quality. However, since then, new experimental results have become publicly available, including RAD21 (Accession: GSM5983424) and CTCF (Accession: GSM5983423) GM12878 ChIP-seq datasets from the Ruan Lab (PMID: 38585764).

In response to your suggestion, we have now downloaded the peaks from the GM12878 RAD21 ChIP-seq dataset (GEO accession GSM5983424) and assessed the overlap with both the SMC3 and SA1 ChIP-seq datasets. Our findings (presented in Figure R1, bottom panels) indicate that the RAD21 ChIP-seq data has a smaller overlap with both the SA1 and SMC3 datasets, recovering 48% of SA1 peaks and 65% of the SMC3 peaks. This suggests that, while valuable, the RAD21 dataset may not provide as comprehensive coverage of cohesin binding sites as the SA1 dataset used in our study.

Given these considerations, we believe that the SA1 ChIP-seq dataset remains the most suitable choice for our analysis. Nonetheless, we acknowledge the potential value of the RAD21 dataset and will consider incorporating it into future research projects to further enhance our understanding of cohesin dynamics.

Figure R1. Overlap of peaks from different GM12878 cohesin ChIP-seq datasets. GM12878 SA1 ChIP-seq (Kasowski et al., 2013), GM12878 SMC3 ChIP-seq (Zhang et al. 2020), GM12878 RAD21 ChIP-seq (Kim et al. 2024)

References -

- Kasowski, M., Kyriazopoulou-Panagiotopoulou, S., Grubert, F., Zaugg, J. B., Kundaje, A., Liu, Y., Boyle, A. P., Zhang, Q. C., Zakharia, F., Spacek, D. V., Li, J., Xie, D., Olarerin-George, A., Steinmetz, L. M., Hogenesch, J. B., Kellis, M., Batzoglu, S., & Snyder, M. (2013). Extensive variation in chromatin states across humans. *Science (New York, N.Y.)*, 342(6159), 750–752.

<https://doi.org/10.1126/science.1242510>

- Zhang, J., Lee, D., Dhiman, V., Jiang, P., Xu, J., McGillivray, P., Yang, H., Liu, J., Meyerson, W., Clarke, D., Gu, M., Li, S., Lou, S., Xu, J., Lochovsky, L., Ung, M., Ma, L., Yu, S., Cao, Q., Harmanci, A., ... Gerstein, M. (2020). An integrative ENCODE resource for cancer genomics. *Nature communications*, 11(1), 3696. <https://doi.org/10.1038/s41467-020-14743-w>
- Kim, M., Wang, P., Clow, P. A., Chien, I. E., Wang, X., Peng, J., Chai, H., Liu, X., Lee, B., Ngan, C. Y., Yue, F., Milenkovic, O., Chuang, J. H., Wei, C. L., Casellas, R., Cheng, A. W., & Ruan, Y. (2024). Multifaceted roles of cohesin in regulating transcriptional loops. *bioRxiv* : the preprint server for biology, 2024.03.25.586715. <https://doi.org/10.1101/2024.03.25.586715>

2. The authors state in the second paragraph on page 13 that “ChIP-seq experiments for HG00731 were not available”. Theoretically, since the HiChIP process involves immunoprecipitating the target protein after the standard Hi-C ligation step, if a HiChIP experiment can be performed, the corresponding ChIP-seq experiment should be relatively easy to conduct.

R2: We agree with the accurate comment of the reviewer. As the HiChIP experiment consists of HiC and ChIP-seq steps, and both parts of the procedure had already been optimised, the generation of a corresponding ChIP-seq experiment in principle is relatively simple. Unfortunately, due to financial constraints, we were unable to generate additional experiments for this particular project. Nevertheless, we focused on performing a thorough bioinformatic analysis on our own and publicly available data (Fig. 1) that proves the validity of our approach.

Analytical approach proposed in the manuscript is especially advantageous for projects where performing additional ChIP-seq experiments involves significant financial and labour costs. For example, studies focused on examining the differences in 3D chromatin structure between many individuals (e.g. three family trios: mother, father and child from the 1000 Genome Project) or under many conditions in parallel (e.g. use of different sources of cellular stress). In such an experimental setup, each additional experiment such as ChIP-seq, which must be performed in at least two replicates for every condition/sample, significantly increases the costs of the project. This is particularly important when the biological model of interest has no publicly available ChIP-seq data. Therefore, the aim of our work was to show that if the HiChIP experiment presents a high signal-to-noise ratio, additional ChIP-seq can be omitted. Nevertheless, it is important to keep in mind that for HiChIP data presenting poorer signal-to-noise ratio this approach might not be optimal and additional ChIP-seq experiment may be required.

- To address this topic in the manuscript we decided to add the following fragment to the 4th paragraph of the discussion section:

“Our results suggest that if the HiChIP experiment has a high signal-to-noise ratio, additional ChIP-seq is not required for subsequent data analysis. Such an approach may be particularly advantageous for projects investigating differences in 3D chromatin structure between many individuals or under various experimental treatment conditions in parallel (e.g. use of different sources of cellular stress). In such an experimental setup, each additional high-throughput experiment such as ChIP-seq, which must be performed in at least two replicates for each condition/sample, significantly increases the financial and labour costs of the project. However, it is important to note that for HiChIP data with poorer signal-to-noise ratio, this strategy may not be optimal and an additional ChIP-seq experiment may be required. Importantly, our pipeline also supports the integration of ChIP-seq data in such cases.”

3. It was noted that over 600,000 peaks were obtained from the cohesin HiChIP data, many of which do not show a clear distinction from the background. The source article for this data (Mumbach, 2016) identified approximately 28,000 peaks, whether the parameters chosen for peak calling were appropriate?

R3. We applied identical parameters for all analysed datasets, including both ChIP-seq and HiChIP, to maintain consistency. However, it is important to note that the datasets from the original study (Mumbach, 2016) were pre-processed using a different pipeline. Specifically, the authors utilised the HiC-Pro pipeline, which processes HiChIP data differently by combining dangling-end and self-ligation reads to generate a compatible input for peak calling with MACS2. In contrast, we followed the standard processing workflow for paired-end ChIP-seq experiments, including read mapping and processing.

For peak calling, we used MACS3 with the 'no model' option and an FDR cutoff of 0.01 (1%), consistent with standard practices. However, unlike the original study, we did not manually set the 'extsize' parameter and instead used the default value of 200, as recommended for CTCF in the MACS3 documentation. This variation in parameterization may account for the difference in the number of peaks identified.

To address the reviewer's concern, we repeated the peak calling for the GM12878 SMC1 HiChIP sample, this time specifying the 'extsize' parameter as 147, consistent with the original study. This adjustment resulted in 100,411 peaks for the GM12878 SMC1 HiChIP sample, compared to the over 600,000 peaks initially identified. To further evaluate the impact of this parameter change, we also applied it to the HG00731 SMC1 dcHiChIP sample which led to modest increase in the peak count by approximately 3% (from ~97,000 to 100,000 peaks).

Upon visual inspection of the newly identified peaks for our sample, we observed that many of the additional peaks appeared to be very low (close to the background), indicating that the increase in peak numbers due to the parameter adjustment does not necessarily improve the biological relevance of the results. Therefore, while adjusting the 'extsize' parameter reduces the number of peaks, it does not significantly enhance the quality of the peak calls in our dataset. This suggests that the difference in peak counts between our analysis and the source article is primarily due to differences in data processing pipelines and parameter settings.

- Therefore align with the above explanation we have added the following text in the "Data Processing and Analysis", subsection "nf-HiChIP pipeline", 2nd paragraph -

with the 'no model' option and using default value of 200 for 'extsize' parameter for all the HiChIP and ChIP-seq samples

4. In the nf-HiChIP pipeline, HiChIP data was used instead of ChIP-seq data to obtain signal peaks and subsequently annotate loops. Testing with public data revealed that high signal-to-noise ratio data is necessary. For data with a moderate signal-to-noise ratio (e.g., GSM2774002, HiChIP_K562_YY1), the pipeline reported errors and stopped running. Additionally, it was noted that the pipeline does not support stepwise execution; if an error occurs at a specific step or if the code is interrupted, it cannot resume from that step and must be restarted from the beginning

R4. Thank you for your valuable feedback and for bringing these important points to our attention.

Regarding the YY1 HiChIP sample presenting (GSM2774002, HiChIP_K562_YY1) moderate signal-to-noise ratio:

Our pipeline is indeed optimised for samples that produce distinct peaks that can be reliably called using a ChIP-seq-like approach involving specific mapping and initial data filtering procedures. We tested the YY1 HiChIP sample and were able to reproduce the errors. Upon investigation, we found that the low signal-to-noise ratio of this data prevented efficient peak calling. Peak calling using MACS3 identified only 126 peaks. The subsequent errors in the nf-HiChIP pipeline were due to this insufficient number of peaks. To address this issue, we ran the nf-HiChIP pipeline on the same sample but using an external peak set. Specifically, we used publicly available K562 YY1 ChIP-seq peaks from the ENCODE project (ENCODE ID: ENCFF398UQZ). With this external YY1 peak set, the nf-HiChIP pipeline was successfully completed and returned only 93 loops.

We also noted that the original study reporting this sample (Weintraub et al. 2017) did not include peak-dependent steps in their analysis. Instead, the authors identified all interactions and then overlapped them with publicly available K562 YY1 ChIP-seq peaks. To verify this approach, we performed an additional analysis using the Juicer pipeline. This analysis identified 1,427 loops, which is significantly fewer than the number of loops identified by Juicer for the HiChIP samples included in our study. This discrepancy may be due to the low sequencing depth of the YY1 HiChIP sample (~31 million reads for the first replicate and ~49 million reads for the second replicate), compared to the deeper sequencing of samples in our study (between 300 and 770 million reads, Supplementary Table 2). Additionally, the low number of interactions may be attributed to the low signal-to-noise ratio observed in the coverage tracks (Figure R2).

- To address this topic in our manuscript, we have added the following text to the fourth paragraph of the Discussion section:

Our results suggest that if the HiChIP experiment has a high signal-to-noise ratio, additional ChIP-seq is not required for subsequent data analysis. Such an approach may be particularly advantageous for projects investigating differences in 3D chromatin structure between many individuals or under various experimental treatment conditions in parallel (e.g. use of different sources of cellular stress). In such an experimental setup, each additional high-throughput experiment such as ChIP-seq, which must be performed in at least two replicates for each condition/sample, significantly increases the financial and labour costs of the project. However, it is important to note that for HiChIP data with poorer signal-to-noise ratio, this strategy may not be optimal and an additional ChIP-seq experiment may be required. Importantly, our pipeline also supports the integration of ChIP-seq data in such cases.

Regarding the pipeline's ability to resume execution:

The nf-HiChIP pipeline does support resuming from the last successfully executed step using the `-resume` flag. This feature is inherent to Nextflow, the workflow management system upon which our pipeline is built. While this functionality is standard in Nextflow and documented in its official documentation (<https://www.nextflow.io/docs/latest/cache.html#resuming-executions>), we recognize that it may not be immediately apparent to all users. We will update our repository's documentation to include a note about the `-resume` flag to assist users who may encounter interruptions during execution.

Once again, we appreciate your constructive feedback, which has helped us improve both our manuscript and our pipeline documentation.

Figure R2. Genome coverage tracks of samples included in our manuscript: HG00731 SMC1 dcHiChIP, HG00731 CTCF HiChIP, GM12878 SMC1 HiChIP (Mumbach et al., 2016), Rec1 SMC1 HiChIP (Petrovic et al., 2019) (blue) and K562 YY1 HiChIP (Weintraub et al., 2017) (red). Compared to other samples K562 YY1 HiChIP seems to have a very low signal-to-noise ratio.

References -

- Mumbach, M. R., Rubin, A. J., Flynn, R. A., Dai, C., Khavari, P. A., Greenleaf, W. J., & Chang, H. Y. (2016). HiChIP: efficient and sensitive analysis of protein-directed genome architecture. *Nature methods*, 13(11), 919-922. <https://doi.org/10.1038/nmeth.3999>
- Petrovic, J., Zhou, Y., Fasolino, M., Goldman, N., Schwartz, G.W., Mumbach, M.R., Nguyen, S.C., Rome, K.S., Sela, Y., Zapataro, Z. and Blacklow, S.C., (2019). Oncogenic notch promotes long-range regulatory interactions within hyperconnected 3D cliques. *Molecular cell*, 73(6), 1174-1190. <http://dx.doi.org/10.1101/527325>
- Weintraub, A.S., Li, C.H., Zamudio, A.V., Sigova, A.A., Hannett, N.M., Day, D.S., Abraham, B.J., Cohen, M.A., Nabet, B., Buckley, D.L. and Guo, Y.E., (2017). YY1 is a structural regulator of enhancer-promoter loops. *Cell*, 171(7), 1573-1588 <https://doi.org/10.1016/j.cell.2017.11.008>

5. In the second paragraph on page 16, line 12, the authors state that approximately 75% to 90% of the nf-HiChIP loops overlap with those detected by HiCCUPS. Given that the manuscript mentions the two methods detected 80,487 and 43,148 loops, respectively, this ratio raises concerns.

R5. Thank you for bringing up this critical point regarding the overlap between nf-HiChIP loops and those detected by HiCCUPS. We appreciate the opportunity to clarify our methodology and address your concerns.

The observed discrepancy arises from fundamental differences in how the two loop identification methods - HiCCUPS and MAPS - detect and define chromatin loops. HiCCUPS employs an image processing based approach to identify loops from chromatin interaction maps, calling loops at three different resolutions: 5 kb, 10 kb, and 25 kb. This multi-resolution strategy affects the size and definition of loop anchors. Additionally, the Juicer pipeline, which includes HiCCUPS, merges nearby

loops based on predefined thresholds, resulting in broader loop anchors that may encompass multiple interaction sites. In our analysis, HiCCUPS identified 43148 loops, reflecting its conservative merging and multi-resolution approach.

In contrast, MAPS detects loops directly from interaction data at a single, user-defined resolution. For our nf-HiChIP analysis, we set the resolution to 5 kb, resulting in loop anchors precisely 5 kb in size. Unlike HiCCUPS, MAPS does not merge overlapping or closely localised loops. Each interaction is treated as a separate loop, even if it is adjacent to others. This approach led to the identification of 80487 loops in our nf-HiChIP analysis.

Due to these methodological differences (particularly the higher resolution and lack of merging in MAPS) multiple nf-HiChIP loops can overlap with a single HiCCUPS loop. When calculating the overlap, we find that approximately 75% to 90% of nf-HiChIP loops correspond to regions identified by HiCCUPS. As illustrated in Figure R3, several nf-HiChIP loops detected by MAPS overlap with a single, broader HiCCUPS loop. This phenomenon explains the high percentage of overlap despite the different total number of loops. The fixed anchor size of 5 kb in nf-HiChIP allows for more precise localisation of interaction sites, potentially identifying sub-loops within the broader loops anchors detected by HiCCUPS.

In conclusion, the overlap ratio reflects the fact that nf-HiChIP (using MAPS) identifies loops at a finer resolution without merging, resulting in a higher total number of loops that nonetheless correspond to the broader loops detected by HiCCUPS. The high percentage of overlap indicates that, despite methodological differences, both approaches capture similar underlying chromatin interactions (Figure R3).

Figure R3. Example of overlap between multiple loops from nf-HiChIP with a single loop from HiCCUPS.

To aid reader understanding of the overlap analysis, we made the following updates:

- In the “Data Processing and Analysis” section, within the “Juicer” subsection, we added the

following sentence to the first paragraph:

HiCCUPS employs an image processing-based approach to identify loops from chromatin interaction maps, calling loops at three different resolutions: 5 kb, 10 kb, and 25 kb. This multi-resolution strategy affects the size and definition of loop anchors. Additionally, the Juicer pipeline, which includes HiCCUPS, merges nearby loops based on predefined thresholds, resulting in broader loop anchors that may encompass multiple interaction sites.

- In the same “Data Processing and Analysis” section, we included a new subsection, “Loop and stripe overlap analysis,” which details the strategy used for overlap analysis across samples.
- Additionally, we added a dedicated figure (Supplementary Figure 5) illustrating the loop overlap analysis.

6. The authors detected loops from cohesin FA-EGS HiChIP data using three different methods: nf-HiChIP, HiCCUPS, and ChIA-PET. If additional data, such as CTCF binding patterns, associations with promoters or enhancers, and types of histone modifications, could be incorporated, it would provide more compelling evidence for evaluating the reliability of loops annotated by different methods based on potential biological relevance.

R6. Thank you for the valuable suggestion. In accordance with your recommendation, we have included a new supplementary table (Supplementary Table 6) that displays the CTCF binding patterns across all samples, as identified by the three different methods: nf-HiChIP, HiCCUPS, and ChIA-PET. This addition provides a clearer understanding of CTCF binding patterns in relation to the significant interactions.

Following your recommendation, we have also annotated the loop into Enhancer-Promoter (EP) and Promoter-Promoter (PP) loop category. We mapped the GM12878 cell specific Enhancer and Promoter identified from ChromHMM chromatin state calls to the loops detected for all analysed cell lines from different methods: nf-HiChIP, HiCCUPS, and ChIA-PET. Statistics for the identified EP and PP loop are provided in the new supplementary table (Supplementary Table 7).

In accordance with this we have added the following sentences -

- In section “Data Processing and Analysis”, after subsection “CTCF motifs”, we added new subsection

Annotation of loops to enhance and promoter

To perform annotation, we downloaded cell-type-specific enhancer and promoter regions from ChromHMM chromatin state calls ⁴⁸ for the GM12878 cell line from ENCODE ⁴⁹. The loop anchors were then separated into right and left anchors. Each anchor file was subsequently intersected with the enhancer and promoter files using Bedtools. Following the intersections, custom scripts were used to categorise the loops as Enhancer-Promoter (EP) or Promoter-Promoter (PP) interactions.

- In section “Result”, after subsection “Dual cross-linking HiChIP protocol improves detection of cohesin-mediated loops”, in 6th paragraph

Chromatin loops were identified and categorized as Enhancer-Promoter (EP) or Promoter-Promoter (PP) across different cell lines using three loop-calling methods

(Supplementary Table 7). Approximately 40%-50% of the loops identified by nf-HiChIP and HiCCUPS were classified as EP or PP, whereas the proportion was slightly lower for ChIA-PIPE (around 20%-35%), likely due to the higher overall number of loops identified by this method. These findings providing deeper insights into the biological relevance of these loops.

7. Since the gStripe algorithm relies on the detected loops, it would be beneficial to provide an evaluation or application standard instead of merely reporting detection rates among different methods. This would help determine the suitability of the loops obtained using various methods, particularly nf-HiChIP and ChIA-PET, for analyzing the intended data.

R7. Thank you for this valuable suggestion. We have expanded the paragraph at the end of the relevant section, discussing the usefulness of different stripe callers for providing data to gStripe. Moreover, in line with Yoon et al., 2022, in the Supplementary Figure 11 we show a comparison of the pileup plots of the stripe regions presented in Figure 3B. These plots illustrate that the stripes observed in the FA-EGS HiChIP (HG00731) contact map are more pronounced than those in HiChIP GM12878, similar to what can be observed for the loops.

It would indeed be beneficial to have a standard method for assessing stripe quality. However, to the best of our knowledge, no such standard currently exists, nor is any stripe calling tool widely accepted as a benchmark. A major challenge in assessing and comparing of stripe quality lies in the fact that the measures used by both Stripenn and gStripe (and, to the best of our knowledge in all stripe calling methods) are based on comparing some property of the candidate stripe with the background established on the same dataset. For instance, the measures: “p-value” (to clarify: an aggregate of p-values of stripe fragments), “stripiness” (for Stripenn) or “stripe score” (gStripe) represent the distinctiveness / strength / quality of the stripe in relation to the rest of that particular heatmap / loop set. When signal characteristics differ between datasets - particularly those generated by different experimental protocols - these values become difficult to compare directly. In practice, ad hoc approaches such as pileup plots are often used, as demonstrated in Yoon et al.

In line with the suggestion, we have added the following sentences to provide a broader understanding of the evaluation of the stripes called by different methods.

- In the Data Processing and Analysis section, subsection “Calling architectural stripes”:

Pileup plots for the stripe analysis were constructed using Coolpuppy⁶⁸ API. In line with the methodology used in Stripenn we used parameters equivalent to “--rescale --local --unbalanced” options. In this procedure each stripe domain is expanded by regions of equal size from both sides, then rescaled to a standard size, and the raw contact matrix signal is averaged over all stripe domains in a given dataset.

- In Result Section, subsection “Dual cross-linking HiChIP protocol reveals architectural stripes in more detail”:

“and constructed pileup plots of the stripe domains in line with the methodology used in Stripenn The plots show the interaction matrix signal averaged over the stripe domains in each dataset (Supplementary Figure 11). Crucially, we observed that the plots for cohesin FA-EGS HiChIP (HG00731) have sharper and more distinct stripe features than those obtained from cohesin FA HiChIP (GM12878), regardless of the stripe calling method. Moving further to visual inspection of individual stripes, we noticed

the pileup plots, where the Stripenn stripes appear as wide, blurry flares, while gStripe stripes are thin and sharp

- In Result Section, subsection “Dual cross-linking HiChIP protocol reveals architectural stripes in more detail” and after paragraph 6 -

Finally, we evaluate the suitability of the three loop callers (nf-HiChIP, HiCCUPS and ChIA-PIPE) for use with gStripe on FA-EGS HiChIP data. While ChIA-PIPE identifies the largest number of stripes, they frequently occupy overlapping domains (86% in the FA-EGS HiChIP sample vs. 57% for nf-HiChIP, see above), which may not always be desirable. In contrast, nf-HiChIP stripes exhibit the highest consistency across datasets, with 51%-75% stripes from GM12878 also present in HG00731, and they show the strongest pile-up plot signal among the gStripe results (Supplementary Figure 11). Thus, while specific research goals may influence the choice of loop calling method, we generally recommend nf-HiChIP for generating input for gStripe, unless densely packed stripes in large numbers are needed, in which case ChIA-PIPE is a viable alternative.

- In line with the suggestion, we have added an additional figure (Supplementary Figure 11) in which pileup plots present averaged contact map values in the stripe domains.

8. The description of data overlap in the manuscript is not clear enough, for example, figure 2D requires consulting the figure legend for proper understanding.

R8. We apologise for the earlier unclear description of the overlap statistics. To address this, we have added a detailed subsection, “Loop and Stripe Overlap Statistics for Heatmaps”, which explains in detail the strategy used to calculate overlap for loops and stripes. This new section describes the methods used to identify overlaps between loop anchors (Figure 2D), including the 15 kb tolerance adjustment, anchor extension, and overlap thresholds. This clarification provides an in-depth look at the computational approach underlying the overlap representation in the heatmaps.

- Therefore in the section “Data Processing and Analysis”, we have added the subsection “Loop and stripes overlap statistics for heatmaps” with the following text changes -

Two loops are considered overlapping if both of their anchors overlap by at least

1 base pair (bp). To account for a 15 kb tolerance in overlap detection, the start position of one loop's anchor is extended 15 kb upstream, and its end position is extended 15 kb downstream. An overlap of at least 1 bp between the adjusted anchor of one loop and the anchor of the other loop is then checked to confirm the overlap. Similarly, two stripes are considered overlapping if they 1) share the same orientation (horizontal or vertical) and 2) their anchor regions (i.e. the 1D regions containing the coordinate where the visible stripe intersects with the diagonal) are no further than 10kb apart. Finally, two striping domains (1D regions containing the whole extent of the stripe) are considered overlapping, if they overlap at least by 1bp after expanding kb in both directions. These calculations are performed using custom in-house scripts and are performed for stripes, and stripe domains between all HiChIP samples. When calculating overlaps between two sets of 1D regions (let us call them A and B), such as loop anchors, stripe anchors, domains etc., one-to-many overlaps are possible (i.e. a situation, where multiple regions from set A intersect one large region in set B, or vice versa). Therefore, to properly quantify the overlaps, we report both the number of elements from set A, for which at least one overlapping region was found in set B, and the reverse - number of elements in set B that were matched to at least one region in set A. We report this

in the form of a matrix, which in the general case would not be symmetrical.

When calculating overlaps between two sets of 1D regions (let us call them A and B), such as loop anchors, stripe anchors, domains etc., one-to-many overlaps are possible (i.e. a situation, where multiple regions from set A intersect one large region in set B, or vice versa). Therefore, to properly quantify the overlaps, we report both the number of elements from set A, for which at least one overlapping region was found in set B, and the reverse - number of elements in set B that were matched to at least one region in set A. We report this in the form of a matrix, which in the general case would not be symmetrical.

- Form the same section, following sentence has been removed -

Since one-to-many overlaps are possible for two sets of regions (anchors, stripes, etc.), the resulting matrices are not symmetric. For two stripes to be considered overlapping we require that they have the same orientation (horizontal or vertical) and their anchors intersect with 10kb tolerance and we do not consider the lengths of the stripes.

9. token or temporary link to access the data was not provided by the authors.

R9. We apologise for the inconvenience, here we provide the link and token for the access to the data files uploaded on the GEO.

To review GEO accession GSE266640:

Go to <https://www.ncbi.nlm.nih.gov/geo/query/acc.cgi?acc=GSE266640>

Enter token **odmbsskerfmhfy** into the box

We would also like to inform you that, along with our submission to NCBI GEO (Accession ID: GSE266640), we have successfully deposited all raw data and processed metadata -including BAM files, BigWig tracks, narrowPeak files, Hi-C matrices, loops, and stripes - for each replicate and the merged data from dcHiChIP SMC1 HG00731 (Accession ID: 4DNES7BZ38JT) and HiChIP CTCF HG00731 (Accession ID: 4DNESOAF3QAA) at the 4D Nucleome Data Portal (<https://data.4dnucleome.org/>).

Minor Comments -

1. there are some issues with the supplementary tables in the PDF version, supplementary table1 is split, and tables 2-7 are missing.

R1. We apologise for this technical error held during the submission process. We will ensure that all the tables are properly reported in the revised manuscript.

2. The order of some supplementary figures in the manuscript does not follow the numbering, for example, Supplementary Figure 1B appears before Supplementary Figure 1A, and Supplementary Figure 6 is not mentioned in the manuscript.

R2. We apologise for the oversight. We have carefully reviewed and reordered the supplementary figures to ensure they are referenced in the correct sequence. Additionally, we have included a citation

for Supplementary Figure 6 (which is now Supplementary Figure 8, in the revised version of the manuscript) in the appropriate section of the revised manuscript.

3. In line 9 of the second paragraph on page 2, it should be “10-nm chromatin fiber”.

R3. We have corrected the indicated error.

4. In line 10, it seems should be “a subset of loop anchors has been shown to act as boundaries of TADs.”

R4. We are sorry for the error. However, this sentence has been removed in response to the second comment from reviewer #1.

5. In line 2 of the second paragraph on page 3, there is a missing a closing parenthesis after “reviewed in 24”.

R5. The suggested correction has been implemented.

6. In the HiChIP experimental method, line 9 on page 5, “resuspending in 150 µl of pre-warmed” seems to be missing SDS and its concentration.

R6. The missing part “0.5% SDS” has been added to the sentence.

7. In the second-to-last line of the second paragraph on page 13, it should be “the latter type of experiment.”

R7. The term “former” has been appropriately replaced with “letter.”

8. In the first line of the last paragraph on page 14, it should be “to verify the influence of dual cross-linking on HiChIP.”

R8. We appreciate your attention to detail. We apologise for not being precise in our description. In this paragraph we focused specifically on the chromatin immunoprecipitation (ChIP) step of the HiChIP protocol. We have corrected the sentence accordingly..

9. In the second paragraph on page 17, line 6, “ChIP-PIPE” should be “ChIA-PIPE”

R9. We have rectified the mistake

Reviewer 3:

1. It would be beneficial to test if the dual cross-linking method is also effective for other low-affinity factors such as YY1. Expanding the method to include factors like Wapl could significantly broaden the method's applicability and deepen the biological insights obtainable.

R1. We would like to thank the reviewer for the comments. We agree that having one HiChIP protocol that is efficient for a wide variety of protein factors involved in the establishment of 3D chromatin structure (especially those with low affinity) would be an extremely valuable tool in the field of 3D genomics. However, in laboratory practice, chromatin immunoprecipitation (ChIP) experiments often need to be optimised specifically for the protein / biological model of interest (Kidder et al., 2011; Yu et al., 2021). Technical aspects of the ChIP experiment such as (1) starting cell number, (2) cross-linking parameters, (3) antibody selection, (4) sonication parameters and (5) buffers composition often need to be tested specifically for the target factor in order to obtain high quality data. The efficiency of the experiment may also depend on the cell line / tissue used. Optimising the protocol for one protein factor does not guarantee experimental success for another, as we experienced with the standard HiChIP protocol of Mumbach et al. 2016, which worked well for CTCF, but not for cohesin. Furthermore, a good signal-to-noise ratio of the ChIP experiment performed separately does not necessarily result in a successful HiChIP experiment (e.g. in the case of cohesin, standard ChIP-seq experiments work well, but the standard cohesin HiChIP showed a low signal-to-noise ratio). One of the motivations for describing our results in technical details in this manuscript was to share with the scientific community our findings that changing the cross-linking conditions can significantly improve the quality of cohesin HiChIP. We know from our own experience, as well as from discussions with fellow colleagues at conferences, that the technical aspects described in our manuscript can provide valuable guidance when struggling with ChIP-based or 3C-based experiments.

Our protocol could potentially help to improve the quality of HiChIP for other factors such as WAPL or YY1, but we cannot guarantee this as it depends on many factors as mentioned above. While we do appreciate that testing our experimental protocol for other factors involved in 3D chromatin structure would be an important point for further research, to cover it in sufficient detail seems slightly out of the scope of this manuscript, which is specifically focused on the improvement of cohesin HiChIP.

References -

- Kidder, B. L., Hu, G., & Zhao, K. (2011). ChIP-Seq: technical considerations for obtaining high-quality data. *Nature immunology*, 12(10), 918–922. <https://doi.org/10.1038/ni.2117>
- Yu, M., Juric, I., Abnousi, A., Hu, M., & Ren, B. (2021). Proximity Ligation-Assisted ChIP-Seq (PLAC-Seq). *Methods in molecular biology* (Clifton, N.J.), 2351, 181–199. https://doi.org/10.1007/978-1-0716-1597-3_10

2. The results section contains redundant descriptions in the first and fourth paragraphs concerning the focus on human lymphoblastoid cell lines (LCLs) from the 1000 Genome project and the assessment of the dual cross-linking approach's quality. Streamlining these descriptions would enhance clarity and reader comprehension.

R2. Following the reviewer's advice, we have shortened and modified the first four paragraphs of the Results section to streamline the text. We believe this has improved the clarity of the description.

3. To detect loops connecting two distant genomic fragments, the study employs three independent algorithms. I recommend including additional algorithms such as hichipper (Lareau and Aryee, 2018) and cLoops2 (Cao et al., 2022) to potentially improve loop detection accuracy. It would also be informative to include the number of loops called for different datasets and by different software in Figures 2C, S4C, and S5C.

R3. We thank the reviewer for their valuable recommendation. In response, we have processed the SMC1 dcHiChIP HG00731 sample using three additional independent algorithms: hichipper (Lareau and Aryee, 2018), FitHiChIP (Bhattacharyya, Sourya et al., 2019), and cLoops2 (Cao et al., 2022). We have also included Supplementary Figure 9, which compares the APA analysis based on loops identified by these different pipelines, as well as updated Supplementary Table 5, detailing the number of significant interactions (loops) detected by each independent algorithm for the SMC1 dcHiChIP HG00731 sample.

We were informed by Reviewer 2 that Supplementary Tables 2-7 were missing from the PDF version of the manuscript. We would like to clarify that Supplementary Table 5, which contains the number of loops identified across different datasets and software, was included in the original submission. We apologise for the inconvenience caused by this technical error and appreciate your understanding.

For the same we have added the following text in the manuscript -

- In the “Results” section and subsection “Dual cross-linking HiChIP protocol improves detection of cohesin-mediated loops”, 4th Paragraph -

To assess the robustness of our results across different loop-calling algorithms, we repeated our analyses using hichipper⁶⁵, FitHiChIP⁶⁶, and cLoops2⁶⁷. We achieved a comparable number of identified interactions (Supplementary Table 5) and APA scores for the cohesin FA-EGS HiChIP experiment (Supplementary Figure 9), indicating the reliability of our method across different loop-calling algorithms.

- In the “Data Processing and Analysis” section, we added 3 more sub-section -

hichipper

HiChIP paired-end reads were mapped to the hg38 reference genome using the HiC-Pro pipeline (version 3.1.0). Default settings were used to align paired reads to identify valid interactions and generate contact maps matrices. Then, HiChIP loops were called using hichipper (version 0.7.7) using valid HiChIP read pairs with the parameter to use pre-identified peaks by nf-HiChIP pipeline.

FitHiChIP

Statistically significant contacts (5kb bin size, max size 2 Mb, min size 10 kb, FDR 0.05) were identified using Hi-C Pro’s allValidPairs file as input for FitHiChIP v11.0. Peaks identified by nf-hichip were used as a reference set of peaks in the FitHiChIP pipeline and default values for the remaining options.

cLoops2

Raw paired-end reads mapped to hg38 were processed using the tracPre2.py script in the

cLoops2 package. Loops were then identified with the *cLoops2* *callLoops* module, using parameters *-eps 200,500,1000, -minPts 10* and *-cut 1000*, requiring a minimum of 3 PETs to support each confident loop.

- We have included the information in the Supplementary Table 5 and we have added the Supplementary Figure 9 for greater clarity.

4. Supplementary Figure 7A is mentioned before Supplementary Figure 6 in the text, and there appears to be a reference missing to Supplementary Figure 6. Adjusting the order and ensuring all figures are correctly referenced would improve the logical flow.

R4. We apologise for the oversight. We have now cited Supplementary Figure 6 (which is now Supplementary Figure 8, in the revised version of the manuscript) in the appropriate section of the revised manuscript. Additionally, we have reviewed and adjusted the order of the supplementary figures to ensure that all figures are properly referenced in the correct sequence, improving the overall logical flow. Thank you for pointing this out.

5. I suggest creating a figure similar to Figure 11 using the *cLoops2* *estRes* to showcase interaction resolution improvements, with data downsampled to the same final read depth for a consistent comparison

R5. We sincerely thank the reviewer for their insightful suggestion to include figures illustrating the estimated reasonable contact matrix resolution (“*estRes*”) generated using *cLoops2*, with data uniformly downsampled to the same final read depth for consistent comparisons. We would like to highlight that all cohesin HiChIP samples analyzed in our study already exhibit comparable read depths, as detailed in Supplementary Table 2. Due to this similarity, the inclusion of an additional figure showcasing *estRes* would likely not yield significant new insights. Moreover, integrating such a figure into the existing structure of the figure panels may disrupt the coherence of the presentation. We kindly seek the reviewer’s understanding in requesting to forgo the inclusion of this specific figure.

Response to the reviewer

Reviewer 3:

1. I still want to see the cLoops2 estRes results of the HiChIP data as a file reply to reviewer only.
Thanks

R1: Thank you for your comments. Here we present the results (Figure R1) from cLoops2 estRes module estimates the resolution of an interaction contact matrix based on singleton PETs and bins for 1kb, 5kb and 10kb.

Figure R1. cLoops2 estRes module for estimating reasonable contact matrix resolution for HG00731 SMC1 data.